# Universal Rates for Interactive Learning

**Steve Hanneke**
Purdue University
steve.hanneke@gmail.com

**Amin Karbasi**
Yale University, Google Research
amin.karbasi@yale.edu

**Shay Moran**
Technion, Google Research
smoran@technion.ac.il

**Grigoris Velegkas**
Yale University
grigoris.velegkas@yale.edu

## Abstract

Consider the task of learning an unknown concept from a given concept class; to what extent does interacting with a domain expert accelerate the learning process? It is common to measure the effectiveness of learning algorithms by plotting the "learning curve", that is, the decay of the error rate as a function of the algorithm's resources (examples, queries, etc). Thus, the overarching question in this work is whether (and which kind of) interaction accelerates the learning curve. Previous work in interactive learning focused on *uniform* bounds on the learning rates which only capture the upper envelope of the learning curves over families of data distributions. We thus formalize our overarching question within the distribution dependent framework of universal learning, which aims to understand the performance of learning algorithms on every data distribution, but without requiring a single upper bound which applies uniformly to all distributions. Our main result reveals a fundamental trichotomy of interactive learning rates, thus providing a complete characterization of universal interactive learning. As a corollary we deduce a strong affirmative answer to our overarching question, showing that interaction is beneficial. Remarkably, we show that in important cases such benefits are realized with label queries, that is, by active learning algorithms. On the other hand, our lower bounds apply to arbitrary binary queries and, hence, they hold in any interactive learning setting.

## 1 Introduction

In supervised learning, arguably the most commonly studied variation of machine learning, we consider a learning algorithm that is typically given access to a training set of $n$ labeled examples, sampled from an unknown distribution. Based on this training set, the goal of the algorithm is to learn to output a concept/function that maps data points to labels so that the probability of making a mistake on unseen (unlabeled) data, drawn from the same distribution, is minimized. However, this passive setting does not capture a lot of applications in which there is an abundance of unlabeled data where labels are not readily available but the learner can interact with the domain expert (such as a human annotator) to acquire more information about the data. Consider, for example, the task of email spam filtering. Service providers, such as Google, have easily access to billions of emails. However, obtaining labels (i.e., spam or not) usually requires a human to read the content of those emails, a process that is quite costly. One way to model such tasks is through the *interactive* learning framework where the learning algorithm is given access to a large (potentially infinite) stream of unlabeled examples from which it can then submit $n$ queries to an expert in order to gain information about these examples. Such queries may include, but are not limited to, asking the labels of those

36th Conference on Neural Information Processing Systems (NeurIPS 2022).

examples (e.g., is this email spam?) or comparing examples with one another (e.g., between these two emails which one looks more like spam?).

In this paper we would like to understand to what extent interacting with an expert who knows the target concept can accelerate the learning process. It is common to measure the effectiveness of passive supervised learning processes by plotting the "learning curve", that is, the decay of the error rate as a function of the number of training examples. Analogously, for interactive learning algorithms one measures the decay of the error rate as a function of the number of queries the algorithm submitted to the domain expert. The setting in which the learner is limited to asking label queries is called *active* learning. The overarching question in this work is whether (and which kinds of) interaction accelerates the learning curve?

Previous work on interactive learning has given rise to a large collection of general theories [KMT93, Das05, BBL09, Han07b, Han07a, DHM07, Han09, YHC10, BH12, Han14, HY15, BHLZ16, KLMZ17] establishing guarantees on the error rate as a function of the number of queries of various types. However, most of these theories are primarily concerned with capturing a *uniform* (i.e., *minimax*) guarantee on the rates of convergence: that is, error bounds holding in the worst case over some family of distributions.[1] Even though these theories give rise to a clean and elegant mathematical framework, they do not fully explain the behavior of learning curves: due to the minimax nature of the guarantees they consider, they can only capture the upper envelope of the learning curves over a family of data distributions. The minimax perspective does not match practical and experimental machine learning, where the target distribution is fixed and the number of training examples and queries varies according to the learner's resources and desired accuracy [CT90, CT92]. In contrast, in order to observe the learning curve that corresponds to uniform/minimax guarantees one would have to vary the data-generating distribution as the constraint on the number of queries the learner can make increases. We therefore formalize our overarching question within the distribution dependent framework of *universal learning* [BHM+21], a framework designed to understand optimal rates of convergence for learning curves. This framework aims to understand the best possible asymptotic rate of convergence that can hold for *every* data distribution, but without requiring an upper bound which applies uniformly to all of these distributions. While there have been a few past works to study universal rates for the special case of *active* learning (i.e., label queries) [BHW08, Han12, YH13], none have yet provided a complete characterization, and none have considered more general forms of interactive learning.

The main result of this paper is a complete characterization of universal interactive learning rates, for learners able to make arbitrary yes/no queries. Specifically, for such learners, we prove there is a fundamental *trichotomy* of possible optimal universal learning rates: any given concept class has an optimal rate decaying at either (i) arbitrarily fast rates, (ii) exponential rates, or (iii) arbitrarily slow rates. Moreover, the optimal rates for classes of type (i) can be attained by learning rules whose interaction is restricted to querying the labels of points in the (unlabeled) input sample. In other words, active learning with arbitrarily fast rates is possible for a variety of classes including, and rather surprisingly, for some classes with infinite VC dimension.

Furthermore, we pinpoint complexity measures to precisely categorize any concept class into one of the above three categories of optimal rate. Indeed, since these same complexity measures were also found by [BHM+21] to characterize a trichotomy of rates for passive supervised learning, our results are directly comparable, and reveal a strong benefit from interactive learning: namely, classes with optimal rate $\frac{1}{n}$ for supervised learning have optimal rate $e^{-n}$ for interactive learning (category ii), while classes with optimal rate $e^{-n}$ for supervised learning have optimal rates that are arbitrarily fast for interactive learning (category i).

In addition to these strong positive results, including for learning with simple label queries, we also note that our lower bounds on the achievable universal rates are quite powerful, as they apply to *any* interactive learning setting with binary-valued responses: for instance, in addition to label queries and general membership queries, they also apply to learners based on *comparison queries* [KLMZ17].

_______________________

[1]To be clear, while many of these theories express distribution-dependent guarantees, the guarantees are typically formulated with the intent of matching some kind of lower bound on the minimax performance over a family of distributions subject to some parameter value (e.g., disagreement coefficient). For instance, most of these results involve a factor based on the VC dimension or covering numbers, both of which could be unbounded, even in the most-favorable scenarios discussed in the present work. As a result, the distribution-dependent guarantees in these works do not capture optimal universal rates.

Finally, novel algorithmic principles arise from our universal interactive learning framework. Recall that passive learning in the distribution-free PAC setting boils down to the Empirical Risk Minimization (ERM) principle via uniform convergence. Analogously, traditional interactive learning gives rise to simple querying policies such as the disagreement-based CAL algorithm [CAL94] algorithm for label queries (which queries the labels of points that are uncertain), or general binary search strategies in a cover of the class [KMT93] in the case of general yes/no queries. Similarly to the ERM principle, these strategies achieve near-optimal distribution-free uniform rates, for their respective types of queries. In contrast, in the universal setting the algorithmic landscape is richer and the design of learning algorithms able to achieve the optimal rates requires great care. Indeed, the learning rules we use to prove our main results rely on techniques only recently introduced to the learning theory literature, such as Gale-Stewart sequential games and adversarial online learning with infinite ordinal-valued Littlestone dimensions [GS53, BHM$^+$21]. Thus, the universal learning setting motivates the development of a richer variety of algorithmic techniques.

**Uniform vs. Universal Rates.** To make the distinction between uniform and universal rates more concrete, we first recall what *uniform* learnability means. (The notation will be introduced formally below.) A concept class $\mathbb{H}$ is uniformly learnable at rate $R(n)$ if there exists a learning rule $\hat{h}_n$ such that

$$\big(\exists C, c > 0\big)\big(\forall \mathrm{P} \in \mathrm{RE}(\mathbb{H})\big) \text{ it holds that } \mathbb{E}[\mathrm{er}(\hat{h}_n)] \leq CR(cn), \forall n \in \mathbb{N}.$$

In words, there exists a learning rule $\hat{h}_n$ and *distribution-independent* constants $C, c > 0$ so that for every realizable data-generating distribution $\mathrm{P} \in \mathrm{RE}(\mathbb{H})$ the expected error of the classifier $\mathbb{E}[\mathrm{er}(\hat{h}_n)]$ is bounded by $CR(cn)$. The definition of *universal* learnability is obtained by a simple rearrangement of the quantifiers of the previous definition. A concept class $\mathbb{H}$ is universally learnable at some rate $R(n)$ if there exists a learning rule $\hat{h}_n$ such that

$$\big(\forall \mathrm{P} \in \mathrm{RE}(\mathbb{H})\big)\big(\exists C, c > 0\big) \text{ such that } \mathbb{E}[\mathrm{er}(\hat{h}_n)] \leq CR(cn), \forall n \in \mathbb{N}.$$

Importantly, this subtle change in the definition allows the constants $C, c$ to be *distribution-dependent*. As is evident from our main result below, this, seemingly minor, change makes the landscape of the admissible learning rates of interactive learning algorithms vastly different.

## 1.1 Main Results

We next present the key definitions and summarize the main results of this work.

The learning problem consists of a *domain* $\mathcal{X}$, where we assume $\mathcal{X}$ to be a Polish space, and a non-empty *concept class* $\mathbb{H} \subseteq \{0,1\}^{\mathcal{X}}$. We assume that $\mathbb{H}$ satisfies standard measurability assumptions (see Appendix A.1).

A *classifier* is a universally measurable function $h : \mathcal{X} \to \{0,1\}$, whose *error rate* is defined to be $\mathrm{er}(h) = \mathrm{er}_{\mathrm{P}}(h) := \mathrm{P}\{(x,y) : h(x) \neq y\}$, where P is a probability distribution over $\mathcal{X} \times \{0,1\}$. We say that P is *realizable* with respect to $\mathbb{H}$ if $\inf_{h \in \mathbb{H}} \mathrm{er}(h) = 0$. We denote with $\mathrm{P}_{\mathcal{X}}$ its marginal distribution on $\mathcal{X}$.

We define an *interactive learning algorithm* to be a sequence of universally measurable functions which, given access to an infinite stream of *unlabeled* data points from $\mathcal{X}$ that are drawn i.i.d. from $\mathrm{P}_{\mathcal{X}}$ and a query budget $n$, output a classifier $\hat{h}_n : \mathcal{X} \times \{0,1\}$; we restrict the algorithm to depend on only a finite (though unbounded) number of the i.i.d. unlabeled examples. The goal of this algorithm is to come up with classifiers whose expected error $\mathbb{E}[\mathrm{er}(\hat{h}_n)]$ decays as fast as possible as a function of the number of queries $n$.

The aim of this paper is to characterize the learning rates achievable by a learning algorithm in a general *interactive* setting, where the algorithm can ask *arbitrary binary-valued queries*: i.e., any yes/no questions. These include, but are not limited to, label queries and comparison queries for data points, membership queries for any $x \in \mathcal{X}$, or any other query having a binary answer. The formal abstraction for this notion of query is provided in Appendix A.1.

We now formalize the notion of achievable learning rates in the universal learning model [BHM$^+$21].

**Definition 1.** *[BHM$^+$21] Fix a concept class $\mathbb{H}$, and let $R : \mathbb{N} \to [0,1], R(n) \xrightarrow{n \to \infty} 0$ be a rate function.*

- $\mathbb{H}$ *is learnable at rate $R$ if there is a learning algorithm $\hat{h}_n$ such that for every realizable distribution $\mathrm{P}$, there exist distribution-dependent $c, C > 0$, for which $\mathbb{E}[\mathrm{er}(\hat{h}_n)] \leq CR(cn), \forall n \in \mathbb{N}$.*

- $\mathbb{H}$ *is not learnable at rate faster than $R$ if for all learning algorithms $\hat{h}_n$ there exists a realizable distribution $\mathrm{P}$ and $c, C > 0$, for which $\mathbb{E}[\mathrm{er}(\hat{h}_n)] \geq CR(cn)$, for infinitely many $n \in \mathbb{N}$.*

- $\mathbb{H}$ *is learnable with optimal rate $R$ if it is learnable at rate $R$ and it is not learnable at rate faster than $R$.*

- $\mathbb{H}$ *admits arbitrarily fast rates if for all rate functions $R$, it is learnable at rate $R$.*

- $\mathbb{H}$ *requires arbitrarily slow rates if for all rate functions $R$, it is not learnable at rate faster than $R$.*

It is known that, unlike the PAC model, in the universal learning setting *every* concept class $\mathbb{H}$ is learnable [Han21, HKSW21, BHM$^+$21]. Nevertheless, $\mathbb{H}$ might require arbitrarily slow learning rates in some cases [DGL96]. While most of the above definitions parallel the definitions from the work of [BHM$^+$21] on supervised learning, the case of arbitrarily *fast* rates is new to the present work. While technically it would also arise in passive learning in the trivial cases of $|\mathbb{H}| = 1$ or $\mathbb{H} = \{h, 1-h\}$, it arises as a highly non-trivial and important case in interactive learning, with many interesting classes, even of infinite cardinality. Moreover, our proofs establishing arbitrarily fast rates develop a novel active learning technique (based purely on label queries).

We are now ready to state one of the main results of this work: a fundamental *trichotomy* of optimal learning rates. That is, we show there are three possible optimal learning rates that a class $\mathbb{H}$ can admit in the interactive learning setting.

**Theorem 1.** *For every concept class $\mathbb{H}$, exactly one of the following cases holds.*

- $\mathbb{H}$ *is interactively learnable with arbitrarily fast rates.*

- $\mathbb{H}$ *is interactively learnable at an optimal rate $e^{-n}$.*

- $\mathbb{H}$ *requires arbitrarily slow rates for interactive learning.*

Our next result characterizes exactly when these rates occur by specifying combinatorial measures of $\mathbb{H}$ that give rise to each one of these cases. Before we state this result, we need to discuss these combinatorial measures. We refer the reader to Appendix A.2 for the formal definitions. We first describe infinite *Littlestone trees*, which were introduced in [BHM$^+$21]. (Finite versions were studied previously by [Lit88]).

**Definition 2** (Informal (see Definition 5)). *A Littlestone tree for $\mathbb{H} \subseteq \{0,1\}^{\mathcal{X}}$ is a complete binary tree of depth $d \leq \infty$ whose nodes are labeled by elements of $\mathcal{X}$ and the edges to the left, right child are labeled by $0, 1$, respectively. We require that for every level $0 \leq n < d$ and every path from the root to a node at level $n + 1$ there is some $h \in \mathbb{H}$ that realizes this path. We say that $\mathbb{H}$ has an infinite Littlestone tree if it has a Littlestone tree of depth $d = \infty$.*

For some intuition, we refer the reader to Figure 1a. We remark that this notion is closely related to the *Littlestone dimension* of $\mathbb{H}$. The Littlestone dimension is defined to be the largest $d \in \mathbb{N}$ such that $\mathbb{H}$ has a Littlestone tree of depth $d$ and it is $\infty$ if there are Littlestone trees of arbitrary depth. However, this does not necessarily imply that there is a *single* tree whose depth is infinite, so having infinite Littlestone dimension is *not* the same as having an infinite Littlestone tree.

We next discuss the notion of a VCL tree introduced by [BHM$^+$21].

**Definition 3** (Informal (see Definition 6)). *A VCL tree for $\mathbb{H} \subseteq \{0,1\}^{\mathcal{X}}$ is a complete tree of depth $d \leq \infty$ such that every level $0 \leq n < d$ has nodes that are labeled by $\mathcal{X}^{n+1}$ with branching factor $2^{n+1}$ and whose $2^{n+1}$ edges connecting a node to its children are labeled by the elements of $\{0,1\}^{n+1}$. We require that for every node at any finite level $1 \leq n \leq d$, the path from the root to this node is realized by some $h \in \mathbb{H}$. We say that $\mathbb{H}$ has an infinite VCL tree if it has a VCL tree of depth $d = \infty$.*

For a pictorial representation of a VCL tree, we refer the reader to Figure 1b. Intuitively, the difference between a Littlestone tree and a VCL tree is that in the latter, the size of the nodes is increasing

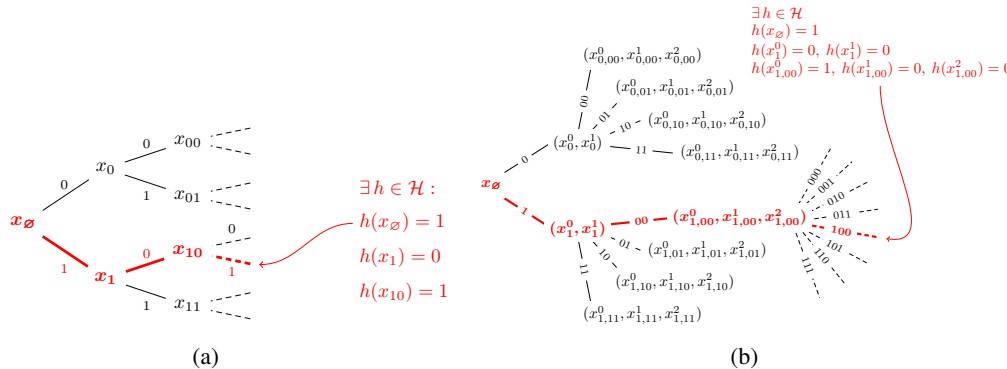

Figure 1: (a) A Littlestone tree of depth 3 is illustrated. (b) A VCL tree of depth 3 is illustrated. Figures are reproduced from [BHM+21] with permission.

linearly with the depth of the tree and the branching factor is increasing exponentially. Recall that in a Littlestone tree both quantities are constant. Intuitively, as we move further down the VCL tree along any path, the class $\mathbb{H}$ needs to be able to shatter sets of increasing size, while respecting label constraints imposed by all previous levels of the path.

We now state our characterization on the optimal rates achievable by interactive learning algorithms.

**Theorem 2.** *For every concept class $\mathbb{H}$ the following hold:*

- *If $\mathbb{H}$ does not have an infinite Littlestone tree, then it is interactively learnable at an arbitrarily fast rate.*

- *If $\mathbb{H}$ has an infinite Littlestone tree but does not have an infinite VCL tree, then it is interactively learnable with optimal rate $e^{-n}$.*

- *If $\mathbb{H}$ has an infinite VCL tree, then it requires arbitrarily slow rates.*

The proof of this theorem is divided into two parts. In Appendix B we provide the bounds relating to Littlestone trees and in Appendix C we prove the bounds relating to VCL trees. It is easy to see that Theorem 2 implies Theorem 1.

We remark that when the learner has access to $P_{\mathcal{X}}$ and not just to an infinite stream of i.i.d. data from it, then we can improve the first rate of the trichotomy.

**Remark 1.** *If the learner has access to the marginal distribution $P_{\mathcal{X}}$ and $\mathbb{H}$ does not have an infinite Littlestone tree, then it can achieve $\mathbb{E}[er(\hat{h}_n)] = 0$, after a finite ($P$-dependent) number $n \in \mathbb{N}$.*

As mentioned above, the algorithm which achieves the arbitrarily fast rates requires only *label* queries, which are, arguably, the simplest type of queries: that is, queries that select any unlabeled example $X_i$ from the i.i.d. stream and request the target concept's label for $X_i$. Hence, this result applies to the traditional active learning setting.

**Remark 2.** *If $\mathbb{H}$ does not have an infinite Littlestone tree, then there exists an algorithm that achieves arbitrarily fast rates using only label queries.*

**Interactive vs. Passive Learning.** We remark that our results can be directly compared to the optimal universal rates for passive supervised learning, as characterized by [BHM+21]. Specifically, [BHM+21] provide the following characterization for the optimal rates achievable by a learner that has access to $n$ labeled examples that are drawn i.i.d. from $P$.

**Theorem 3.** *[BHM+21]* *For every concept class $\mathbb{H}$ (with $|\mathbb{H}| \geq 3$) the following hold:*

- *If $\mathbb{H}$ does not have an infinite Littlestone tree, then it is passively learnable with optimal rate $e^{-n}$*

- *If $\mathbb{H}$ has an infinite Littlestone tree but does not have an infinite VCL tree, then it is passively learnable with optimal rate $\frac{1}{n}$.*

- *If $\mathbb{H}$ has an infinite VCL tree, then it requires arbitrarily slow rates.*

Our results illustrate the power of interactive learning algorithms compared to their passive counterparts. Notably, if $\mathbb{H}$ does not have an infinite Littlestone tree, the rate is improved from exponential to arbitrarily fast; again, this is even achieved by an algorithm based purely on *label* queries. Moreover, using an interactive learning algorithm we achieve an *exponential* improvement in the case where $\mathbb{H}$ does not have an infinite VCL tree but has an infinite Littlestone tree: reducing a $\frac{1}{n}$ rate for passive learning to an $e^{-n}$ rate for interactive learning. Finally, when $\mathbb{H}$ has an infinite VCL tree we cannot achieve any improvement even in the interactive setting.

Another important comparison is between the rates achievable in the universal vs uniform setting. Specifically, results of [KMT93] imply that for interactive learning with arbitrary binary-valued queries, the optimal uniform rate is (i) 0 after a bounded number of queries if $|\mathbb{H}| < \infty$, (ii) $e^{-n}$ when $|\mathbb{H}| = \infty$ but the VC dimension of $\mathbb{H}$ is finite, or (iii) does not converge to 0 when the VC dimension of $\mathbb{H}$ is infinite. Comparing to our result in Theorem 2, we note that the optimal universal rates allow richer families in each of the cases: that is, the category of any $\mathbb{H}$ in the universal rates trichotomy is never worse than its category in the uniform rates trichotomy, and is often better. For instance, finite VC dimension certainly implies there is no infinite VCL tree, but the opposite is not true, and indeed there are classes of infinite VC dimension which do not even have an infinite Littlestone tree [BHM$^+$21], so that the uniform rates are vaccuous while the universal rates are arbitrarily fast.

## 1.2 Examples

Here we present examples of various hypothesis classes that illustrate the three possible optimal rates.

**Example 1** (Finite classes). *Let $\mathbb{H}$ be a finite hypothesis class over some domain $\mathcal{X}$. It follows immediately that this class has no infinite Littlestone tree. Thus, arbitrarily fast rates are achievable. Recall that for passive learning, the optimal rate was $e^{-n}$ for this scenario. This illustrates the improvement compared to the passive universal learning setting, where the best achievable universal rate is exponential [BHM$^+$21].*

We then consider threshold classifiers over the *natural* numbers.

**Example 2** (Threshold classifiers over $\mathbb{N}$). *Let $\mathcal{X} = \mathbb{N}$ and consider $\mathbb{H} = \{h_t : t \in \mathbb{N}, h_t = \mathbb{1}_{x \geq t}\}$. Notice that even though there are Littlestone trees of arbitrary depth, there is no infinite Littlestone tree [BHM$^+$21]. Hence, it is learnable at an abitrarily fast rate in the interactive setting.*

In the next example we consider threshold classifiers over the *real* numbers.

**Example 3** (Threshold classifiers over $\mathbb{R}$.). *Let $\mathcal{X} = \mathbb{R}$ and $\mathbb{H}$ be the class of all threshold classifiers over the real line, i.e. $\mathbb{H} = \{h_t : t \in \mathbb{R}, h_t = \mathbb{1}_{x \geq t}\}$. This class has an infinite Littlestone tree. However, since its VC dimension is finite, we know there is no infinite VCL tree, and hence the optimal rate is $e^{-n}$.*

Next, an example with infinite VC dimension, which nevertheless is interactively learnable at an arbitrarily fast rate:

**Example 4** (Unions of finite sets.). *Let $\mathcal{X} = \cup_k \mathcal{X}_k$ be the disjoint union of the sets $\mathcal{X}_k$, where $|\mathcal{X}_k| = k$. We also let $\mathbb{H} = \cup_k \mathbb{H}_k$, where $\mathbb{H}_k = \{\mathbb{1}_S : S \subseteq \mathcal{X}_k\}$. The VC dimension of $\mathbb{H}$ is infinite. However, notice that this class does not have an infinite Littlestone tree [BHM$^+$21]. Indeed, let $x \in \mathcal{X}_k$ be the root of the tree. Then, only $h \in \mathbb{H}_k$ can have $h(x) = 1$. Thus, there are finitely many hypotheses consistent with this branch from the root, implying the depth is always limited by the choice of root node. Thus, this class is actively learnable at an arbitrarily fast rate.*

Finally, we present an example that requires arbitrarily slow rates.

**Example 5.** *Let $\mathbb{H}$ be the class of all measurable functions over $\mathcal{X} = \mathbb{N}$. This class admits an infinite VCL tree [BHM$^+$21], hence the learning rate in our setting is arbitrarily slow.*

## 1.3 Techniques

In this section we briefly highlight some techniques that are important for our results. For a more extensive discussion, we refer the reader to Appendix A.3.

**Gale-Stewart Games.** An important tool that our algorithms build upon is the theory of Gale-Stewart games [GS53]. These games consist of two players, a learner $P_L$ and an adversary $P_A$, that interact

over an infinite sequence of discrete timesteps. In every round $t \geq 1$, the adversary presents an element $x_t \in \mathcal{X}_t$ to the learner and, subsequently, she picks an element $y_t \in \mathcal{Y}_t$, where $\mathcal{X}_t, \mathcal{Y}_t$ are some sets. If some predetermined condition is met at some finite round $t \in \mathbb{N}$ (e.g. a set of functions that are consistent with the choices of the players becomes empty), then player $P_L$ wins. If the interactions repeat for an infinite number of steps, then player $P_A$ wins. Thus, we can see that the winning strategy of $P_L$ is *finitely decidable*, which means that the learner knows she has won the game after a *finite* number of steps. The fundamental property that characterizes these games is that either the learner or the adversary has a winning strategy, i.e., playing according to this strategy makes them win the game irrespective of their opponent's moves [GS53, HW+93, Kec12]. In this work, we use Gale-Stewart games where $P_A$ chooses subsets of data and $P_L$ chooses labels for the points, such that the possible play-outs correspond to either Littlestone or VCL trees.

**One-Inclusion Graph Predictor [HLW94].** Another important component that we use is the one-inclusion graph predictor. This is a passive learning algorithm that takes as input $n$ labeled points that are drawn i.i.d. from P and an unlabeled example drawn from $P_\mathcal{X}$ and returns the correct label with probability $1 - O(\mathrm{d}/n)$, where $\mathrm{d}$ is the VC dimension of a given function class $\mathcal{F}$ for which P is realizable; indeed, $\mathcal{F}$ may in fact be any *partial* concept class (see below).

**Partial Concept Classes.** The final building block that we use is a subroutine that interactively learns *partial concept classes*. A partial concept class $\mathcal{F}$ is a set of partial functions $f : \mathcal{X} \to \{0, 1, \star\}$, where $f(x) = \star$ means that $f$ is *undefined* at point $x$. Essentially, partial concepts provide a framework to express *data-dependent* assumptions [AHHM21]. We design an algorithm that achieves exponential rates when learning a partial concept class $\mathcal{F}$ whose VC dimension is bounded (cf. Appendix C.2). In our proofs of the universal rates, this class $\mathcal{F}$ is actually constructed algorithmically given a class $\mathbb{H}$ and the i.i.d. unlabeled data.

## 2 Overview of the Proofs of the Main Result

In this section we provide an outline of the proof of the main results. We refer the reader to Appendix B, C for the full proofs.

Our lower bounds rely on establishing a connection between paths in the infinite tree and a family of possible target concepts. Since the paths can be specified by binary strings, our lower bound results follow from lower bounds we prove for lossy coding, representing a universal variant of rate-distortion theory. We leave the details for the appendix, and spend the remainder of this section on outlining the upper bounds, to focus on the novel algorithmic aspects of this work.

**Arbitrarily Fast Rates.** We first design an algorithm that achieves arbitrarily fast rates whenever $\mathbb{H}$ does not have an infinite Littlestone tree. A key subroutine of our algorithm is the *ordinal Standard Optimal Algorithm* (SOA) [BHM+21]. Essentially, this algorithm generalizes the SOA [Lit88] to handle the case of finite, but not uniformly bounded, Littlestone trees. For a more extensive discussion we kindly refer to Appendix B. Importantly, when this algorithm is executed on a (realizable)[2] sequence of labeled data $(x_1, y_1, x_2, y_2, \ldots)$ it will stop making mistakes after a finite time $t^*$. However, $t^*$ depends on the sequence and there is no bound on how big it is. Our approach that achieves the arbitrarily fast rates is outlined in Figure 2.

The intuition behind our algorithm is the following. If we take a large enough number of unlabeled points $m_1$ and consider all of their $2^{m_1}$ possible classifications, there will be one that is correct. Let $S_1^*$ be the correctly labeled set. For large enough $m_1$, we know that the ordinal SOA trained on $S_1^*$ will output a classifier that is always correct. The issues are that (i) we do not have access to the correct classification, and (ii) we do not know how large $m_1$ needs to be. Since the data are generated by some distribution, the time of the last mistake also follows a distribution which can, potentially, have heavy tails. However, there is some (distribution-dependent) number $m^* \in \mathbb{N}$ for which, when the ordinal SOA is trained on a sequence of size $m^*$, with probability at least $1/2$, it will output a classifier that is correct on *every* point $x \in \mathcal{X}$. Thus, if we consider a lot of batches that have size at least $m^*$ and train the ordinal SOA using these batches, at least $1/3$ of the executions will produce an always correct classifier, with high probability. To overcome the issue that we only have unlabeled data, we consider all the possible classifications of these batches and run the ordinal SOA on each one of them. Moreover, we know that all the correct classifiers produce the same

---

[2]This means that for every $t \geq 1$, there is some $h_t \in \mathbb{H}$ such that $h_t(x_\tau) = y_\tau, 1 \leq \tau \leq t$.

---

**ArbitrarilyFastRates**: Input is a unlabeled data sequence $x_1, x_2, \ldots$, and a query budget $n$.

1. Create sets $S_1 = x_1, \ldots, x_{m_1}, S_2 = x_{m_1+1}, \ldots, x_{m_2}$.
2. Split $S_1$ into $\lfloor \sqrt{m_1} \rfloor$ batches of size $\lfloor \sqrt{m_1} \rfloor$ and consider all the labeled prefixes of the batches in lexicographical order: $b_{i,j}$ denotes the $i$-th batch with the labeled prefix $j$, $1 \leq j \leq 2^{\lfloor \sqrt{m_1} \rfloor + 1}$.
3. Let $\mathbb{A}_0(\cdot; b_{i,j})$ be the output of the ordinal SOA trained on $b_{i,j}$ and denote $g_{i,j}(x) = \mathbb{A}_0(x; b_{i,j})$.
4. Evaluate the classifiers $g_{i,j}(\cdot)$ on the points in $S_2$.
5. Define a set $\mathbb{F}$ of equivalence classes: $g_{i,j}$ and $g_{i',j'}$ are in the same class iff they classify $S_2$ the same.
6. For each $F \in \mathbb{F}$, define

$$\mathrm{rank}(F) = \min \left\{ r : \exists \, 1 \leq i_1 < i_2 \ldots < i_{\lfloor \sqrt{m_1} \rfloor /3} \leq \lfloor \sqrt{m_1} \rfloor, k_1, \ldots, k_{\lfloor \sqrt{m_1} \rfloor /3} \in [2^{r+1}] \right.$$

$$\left. \text{such that } g_{i_j, k_j} \in F, 1 \leq j \leq \lfloor \sqrt{m_1} \rfloor /3 \right\},$$

   or $\mathrm{rank}(F) = \infty$ if no such $r$ exists.
7. Enumerate $\mathbb{F} = \{\mathcal{F}_1, \mathcal{F}_2, \ldots\}$ so that $\mathrm{rank}(\mathcal{F}_\ell)$ is non-decreasing in $\ell$, and pick any $f_\ell \in \mathcal{F}_\ell$ for each $\ell$.
8. For each distinct $i, j \leq \lfloor \sqrt{n} \rfloor$ query the label of any point in $S_2$ on which $f_i, f_j$ disagree.
9. If there is some $f_{\hat{\ell}}$ which is correct on all the points we queried, return $f_{\hat{\ell}}$.
10. Otherwise, return any $f_\ell$.

---

Figure 2: Arbitrarily Fast Rates Algorithm

output when executed on a new set of unlabeled data. This is why we use points from $S_2$ and create equivalence classes of classifiers that have the same output on $S_2$; this guarantees that all the correct classifiers will be in the same class $\mathcal{F}_{\ell^*}$, with probability one. By taking $S_2$ to be large enough, we can make the probability that two classifiers which are in the same equivalence class differ on some $x \in \mathcal{X}$ arbitrarily small. Thus, it suffices to consider one representative $f_\ell$ from every equivalence class $\mathcal{F}_\ell$. Finally, we show that there is an ordering of these representatives which guarantees that, with arbitrarily high probability, the representative $f_{\ell^*}$ of the class $\mathcal{F}_{\ell^*}$ (which the correct classifiers belong to) has ranking at most $i^*$, where $i^*$ is a distribution-dependent constant. Thus, by requesting the labels of points on which pairs $f_i, f_j, 1 \leq i, j \leq \sqrt{n}$, disagree, we can just output the one that is correct on all these points, and if $\lfloor \sqrt{n} \rfloor \geq i^*$, such an $f_i$ will exist and will be $f_{\ell^*}$, with arbitrarily high probability.

We emphasize that this algorithm uses only label queries, so it also applies to the traditional active learning setting. Moreover, if the learner has access to the marginal distribution $P_{\mathcal{X}}$, then all the steps in the algorithm can be implemented exactly, i.e. with probability one, so the output will be a classifier that has zero error rate with probability one.

**Exponential Rates.** We now move to the algorithm that achieves exponential rates whenever $\mathbb{H}$ does not have an infinite VCL tree. As we discussed before, the fact that $\mathbb{H}$ does not have an infinite Littlestone tree gives rise to an algorithm that makes a *finite* number of mistakes on any realizable sequence $(x_1, y_1, \ldots)$. In the case of finite VCL trees, [BHM+21] provide an algorithm, which given such a sequence, learns after a finite number of steps to rule out *patterns in the data*. This means that, for some $n^* \in \mathbb{N}$ there is a function $g : \mathcal{X}^{n^*} \to \{0,1\}^{n^*}$ such that $g(x_1, \ldots, x_{n^*})$ are *not* the correct labels of the points $(x_1, \ldots, x_{n^*})$. [BHM+21] obtain this function by defining an appropriate Gale-Stewart game, called the *VCL game*. In every round $t \geq 1$ of this game, the adversary selects a tuple $(x_t^0, x_t^1, \ldots, x_t^{t-1}) \in \mathcal{X}^t$ of $t$ unlabeled points and the learner picks their labels $(y_t^0, y_t^1, \ldots, y_t^{t-1}) \in \{0,1\}^t$. The goal of the learner is to make the set of concepts from $\mathbb{H}$ that are consistent with the execution of the game empty. [BHM+21] show that if $\mathbb{H}$ does not have an infinite VCL tree, then the learner has a winning strategy and this strategy gives rise to the pattern avoidance function $g$ we described earlier. Importantly, the pattern avoidance function helps us define a *partial* concept class whose VC dimension is *finite*. For details, we refer to Appendix C.1.

The first step in our algorithm, presented in Figure 4, is to use half of the query budget to find the labels of $\lfloor n/2 \rfloor$ points in order to simulate the VCL game. The eventually correct pattern avoidance function $g$ we described before, induces some *data-dependent* constraints. To make the idea of our approach easier to grasp, let us assume that we have access to both the function $g$ and to the number $n^*$. Then, the next step is to consider the partial concept class

$$\mathcal{F}^* = \{f : \mathcal{X} \to \{0, 1, \star\} : \forall (x_1, \ldots, x_{n^*}) \in \mathcal{X}^{n^*}, (f(x_1), \ldots, f(x_{n^*})) \neq g(x_1, \ldots, x_{n^*})\}.$$

Importantly, the VC dimension of $\mathcal{F}^*$ is bounded by $n^* - 1$, since it does not shatter any sequence of length $n^*$. Thus, we have reduced the problem to interactively learning a partial concept class whose VC dimension is bounded. We provide such an algorithm in Figure 3.

---

**PartialInteractiveLearning**: Input is an unlabeled data stream $\{x_1, \ldots, \}$ and a query budget $n$.

1. Take a large enough sample of unlabeled data $S = \{x_1, \ldots, x_m\}$.
2. Let $\hat{\mathbb{H}}|_S$ to be the total concept class of the functions in $\mathcal{F}$ that are not undefined on $S$, i.e.,

$$\hat{\mathbb{H}}|_S = \{f \in \mathcal{F} : f(x) \neq \star, \forall x \in S\},$$

   whose domain is restricted to be $S$.     `## this is a total class with`
   `VC dimension bounded by d.`
3. If $\hat{\mathbb{H}} = \emptyset$, return an arbitrary classifier (e.g. the all-zero classifier).
4. Consider the set $\hat{S}$ of all the $O(m^{\mathrm{d}})$ possible classifications of $S$.
5. Since the learner is allowed to use arbitrary binary queries, it can figure out the labels of all points using $O(\mathrm{d} \log m)$ queries as follows:
   - Divide $\hat{S}$ into two sets of equal size $\hat{S}_1, \hat{S}_2$ and ask whether the true classification is in $\hat{S}_1$. Based on the answer of the query, recurse to the appropriate set.
6. The last step is to use these $m$ labeled points as the training set for the one-inclusion graph supervised learning method of [HLW94] for the class $\mathcal{F}$.

---

Figure 3: Interactive Learning with Partial Concept Classes

The idea of the approach is to take a large enough sample $S$ of $m$ unlabeled points and consider the total concept class $\mathbb{H}^* \subseteq \mathcal{F}^*$ of the functions that are *not* undefined on $S$, whose domain restricted is $S$. Then, $\mathbb{H}^*$ also has VC dimension that is bounded by $n^* - 1$, so there are at most $O(m^{n^*})$ possible classifications of these points. The next step is to find the correct classification of these points using $O(n^* \log m)$ binary queries in total. Finally, using these exponentially many labeled points as the training set for the one-inclusion graph algorithm (cf. Theorem 5), we are able to get a classifier whose error rate decreases exponentially fast.

An important difficulty we need to overcome is that we do not know how long we need to run the VCL game for in order to get a correct pattern avoidance function $g$. Our approach, described in Figure 5, is to use $N = \lfloor n/2 \rfloor$ of our query budget to request the labels of some points in order to run the VCL game on. Then, utilizing a result from [BHM+21], we use $\lfloor N/2 \rfloor$ of the data in order to obtain some estimators $\hat{t}_N$ such that running the game on $\hat{t}_N$ points produces a correct $g$ with probability at least $5/8$ (cf. Lemma 5). Having obtained these estimators, we split the remaining $\lfloor N/2 \rfloor$ into $\hat{N} = \lfloor N/(2\hat{t}_N) \rfloor$ batches of size $\hat{t}_N$. Subsequently, we run the VCL game on each batch and obtain the functions $\hat{y}_{\hat{N}}^i, 1 \leq i \leq \lfloor N/(2\hat{t}_N) \rfloor$. The choice of $\hat{t}_n$ guarantees that, with high probability, at least $9/16$ of these functions will be correct pattern avoidance functions. Then, we consider the partial concept classes $\mathcal{F}_i$ that are induced by these functions, in a similar manner as we described before. Finally, we aggregate all these classes into a *majority class*

$$\mathcal{F}_m^{\hat{N}} = \left\{ f : \mathcal{X} \to \{0, 1, \star\} : \forall \ell \in \mathbb{N}, \forall (x_1, \ldots, x_\ell) \in \mathcal{X}^\ell, \exists \text{ at least } (9/16)\hat{N} \text{ classes } \mathcal{F}_j, 1 \leq j \leq \hat{N} \text{ s.t.} \right.$$

$$\left. \exists \hat{f} \in \mathcal{F}_j \text{ with } (\hat{f}(x_1), \ldots, \hat{f}(x_\ell)) = (f(x_1), \ldots, f(x_\ell)) \right\}.$$

The motivation behind aggregating the classes is that (i) we can show that $\mathcal{F}_m^{\hat{N}}$ has *bounded* VC dimension, and (ii) if $\mathcal{F}_m^{\hat{N}}$ cannot produce a labeling for an unlabeled tuple, then this is not the correct labeling (cf. Lemma 7). An equivalent way to describe $\mathcal{F}_m^{\hat{N}}$ is through a sequence of universally-measurable functions $\left\{ G_\ell : (\mathcal{X} \times \{0,1\})^\ell \to \{0,1\} \right\}_{\ell \in \mathbb{N}}$, where

$$G_\ell(x_1, y_1, \ldots, x_\ell, y_\ell) = \mathbb{1} \left\{ \exists f \in \mathcal{F}_m^{\hat{N}} : (f(x_1), \ldots, f(x_\ell)) = (y_1, \ldots, y_\ell) \right\}.$$



**ExponentialRates**: Input is an unlabeled data stream $\{x_1, \ldots, \}$ and a query budget $n$.

1. Use $\lfloor n/2 \rfloor$ of the query budget to get the labels of $\{x_1, \ldots, x_{\lfloor n/2 \rfloor}\}$.
2. Call the GSubroutine (cf. Figure 5) with points $\{(x_1, y_1), \ldots, (x_{\lfloor n/2 \rfloor}, y_{\lfloor n/2 \rfloor})\}$.
3. Create the partial concept class

$$\mathcal{F}_G = \{f : \mathcal{X} \to \{0, 1, \star\} : \forall \ell \in \mathbb{N}, \forall (x_1, \ldots, x_\ell) \in \mathcal{X}^\ell, G_\ell(x_1, f(x_1), \ldots, x_\ell, f(x_\ell)) = 1\}.$$

4. Run Algorithm 3 on the partial class $\mathcal{F}_G$ with budget $\lfloor n/2 \rfloor$ and return its output.



Figure 4: Exponential Rates Algorithm



**GSubroutine**: Input is a labeled data sequence $\{(x_1, y_1), \ldots, (x_N, y_N)\}$.

1. Use $\lfloor N/2 \rfloor$ of the data to estimate $\hat{t}_N$ (cf. Lemma 5). Let $\hat{N} = \lfloor N/2\hat{t}_N \rfloor$.
2. Use the remaining $\lfloor N/2 \rfloor$ of the data to run the VCL game (cf. Algorithm C.1) and obtain $\hat{\boldsymbol{y}}_{\hat{t}_N}^i, 1 \le i \le \hat{N}$.
3. Estimate for $1 \le i \le \hat{N}$

$$\mathcal{F}_i = \left\{ f : \mathcal{X} \to \{0, 1, *\} : \forall (x_1, \ldots, x_{\tau_{\hat{t}_N}}) \in \mathcal{X}^{\tau_{\hat{t}_N}}, (f(x_1), \ldots, f(x_{\tau_{\hat{t}_N}})) \ne \hat{\boldsymbol{y}}_{\hat{t}_N}^i(x_1, \ldots, x_{\tau_{\hat{t}_N}}) \right\}.$$

4. Estimate the $9/16-$majority class

$$\mathcal{F}_m^{\hat{N}} = \left\{ f : \mathcal{X} \to \{0, 1, \star\} : \forall \ell \in \mathbb{N}, \forall (x_1, \ldots, x_\ell) \in \mathcal{X}^\ell, \exists \text{ at least } (9/16)\hat{N} \text{ classes } \mathcal{F}_j, 1 \le j \le \hat{N} \text{ s.t.} \right.$$

$$\left. \exists \hat{f} \in \mathcal{F}_j \text{ with } (\hat{f}(x_1), \ldots, \hat{f}(x_\ell)) = (f(x_1), \ldots, f(x_\ell)) \right\}.$$

5. Return $\left\{ G_\ell : (\mathcal{X} \times \{0, 1\})^\ell \to \{0, 1\} \right\}_{\ell \in \mathbb{N}}$ where

$$G_\ell(x_1, y_1, \ldots, x_\ell, y_\ell) = \mathbb{1}\left\{ \exists f \in \mathcal{F}_m^{\hat{N}} : (f(x_1), \ldots, f(x_\ell)) = (y_1, \ldots, y_\ell) \right\}.$$



Figure 5: Aggregate Function Subroutine

This leads us to the last step in our algorithm. So far, we have produced a partial concept class $\mathcal{F}_m^{\hat{N}}$ with bounded VC dimension, that produces the correct labelings for any (finite) tuple. Thus, it suffices to use the interactive learning algorithm with the $\lfloor n/2 \rfloor$ remaining queries that learns partial concept classes at an exponential rate, which we described before (cf. Figure 3).

## 3 Conclusion and Future Directions

In this paper we provide a complete characterization of universal interactive learning and reveal a fundamental trichotomy of interactive learning rates. Moreover, we specify exactly which properties of the hypothesis class give rise to each case in the trichotomy. The general interactive model we consider makes our lower bounds particularly strong. We believe that there are important questions in this line of work that are beyond the scope of our paper and need to be addressed, such as characterizing the unlabeled sample complexity, studying weaker types of interactions such as label queries and comparison queries, and designing efficient and practical algorithms for natural classes.

## Acknowledgments and Disclosure of Funding

Amin Karbasi acknowledges funding in direct support of this work from NSF (IIS-1845032), ONR (N00014- 19-1-2406), and the AI Institute for Learning-Enabled Optimization at Scale (TILOS). Shay Moran is a Robert J. Shillman Fellow; he acknowledges support by ISF grant 1225/20, by BSF grant 2018385, by an Azrieli Faculty Fellowship, by Israel PBC-VATAT, by the Technion Center for Machine Learning and Intelligent Systems (MLIS), and by the the European Union (ERC, GENERALIZATION, 101039692). Views and opinions expressed are however those of the author(s) only and do not necessarily reflect those of the European Union or the European Research Council Executive Agency. Neither the European Union nor the granting authority can be held responsible for them. Grigoris Velegkas is supported by NSF (IIS-1845032), an Onassis Foundation PhD Fellowship and a Bodossaki Foundation PhD Fellowship.

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
