# A Omitted Details from Section 1

## A.1 Formal Definition of Learning Setting

Before we provide the formal definition of our learning model, we briefly discuss some basic notions of measures and probabilities on Polish spaces. More comprehensive treatments of these topics can be found in [Kec12, Coh13]. Our discussion follows [BHM$^+$21].

A *Polish space* is a separable topological space that can be metrized by a complete metric. Examples of such spaces include $\mathbb{R}^n$, any compact metric space, any separable Banach space, etc. Also, Polish spaces have the property that they are closed under any countable (or finite) product or disjoint union.

We move on to the discussion of *universally measurable* functions. Let $\mathfrak{F}$ be the Borel $\sigma$-field on some Polish space $\mathcal{X}$. Give some probability measure $\mu$, we denote by $\mathfrak{F}_\mu$ the completion of $\mathfrak{F}$ under $\mu$, i.e., the collections of all subsets of $\mathcal{X}$ that differ from a Borel set on a set of zero probability (at most). We call a set $B \subseteq \mathcal{X}$ *universally measurable* if $B \in \mathfrak{F}_\mu$ for every probability measure $\mu$. Moreover, we call a function $f : \mathcal{X} \to \mathcal{Y}$ universally measurable if $f^{-1}(B)$ is universally measurable, for any universally measurable set $B$. Importantly, universally measurable sets and functions on Polish spaces are the same as Borel sets, from a probabilistic perspective.

We are now ready to provide the (standard) definition regarding the measurability of a concept class $\mathbb{H}$.

**Definition 4.** *We say that a concept class $\mathbb{H}$ of functions $h : \mathcal{X} \to \{0,1\}$ on a Polish space $\mathcal{X}$ is measurable if there is a Polish space $\Theta$ and a Borel-measurable map $h : \mathcal{X} \times \Theta \to \{0,1\}$ so that $\mathbb{H} = \{h(\theta, \cdot) : \theta \in \Theta\}$.*

We note that the above definition is very general and it only requires that $\mathbb{H}$ can be parameterized in some reasonable way.

To model the types of queries that the learner is allowed to ask, we let $\Theta^*$ be a set such that $\Theta \subseteq \Theta^*$, and we extend the definition of $h(\cdot, \cdot)$ to $\Theta^*$ in a way that guarantees, for every distribution P realizable with respect to $\mathbb{H}$, there is some $\theta^* \in \Theta^*$ with $h(\theta^*, \cdot)$ a measurable function having $\mathrm{er}(h(\theta^*, \cdot)) = 0$. We remark that $\Theta^*$ does *not* need to be structured, e.g. we do not require it to be a Polish space. We also note that such a set $\Theta^*$ always exists (e.g., we could choose $\Theta^*$ to parameterize the set of all measurable binary functions). Then, we allow the learner to ask binary queries about any set $\hat{\Theta} \subseteq \Theta^*$: that is, a general binary-valued query in this context is formally defined by choosing (possibly in a data-dependent way) any $\hat{\Theta} \subseteq \Theta^*$, and the query is answered with 1 if $\theta^* \in \hat{\Theta}$ and otherwise is answered with 0. Importantly, these queries capture previously studied interactive learning settings such as interactive learning with label queries (where $\Theta^*$ may represent the set of all binary functions, and a query $\hat{\Theta}$ may represent the set of all such functions $h$ with $h(x) = 1$ for a given data point $x$), and interactive learning with comparison queries (where $\Theta^*$ may represent a set of possible real-valued functions, and a query $\hat{\Theta}$ may represent the set of all such functions $f$ such that $f(x) \geq f(x')$ for a given pair of data points $x, x'$).

## A.2 Omitted Definitions

**Definition 5** (Littlestone Tree [BHM$^+$21]). *A Littlestone tree for $\mathbb{H} \subseteq \{0,1\}^{\mathcal{X}}$ is a complete binary tree of depth $d \leq \infty$ whose internal nodes are labeled by $\mathcal{X}$, and whose two edges connecting a node to its children are labeled by $\{0,1\}$, such that every path of length at most $d$ emanating from the root is consistent with a concept $h \in \mathbb{H}$. Formally, a Littlestone tree is a collection*

$$\bigcup_{0 \leq \ell < d} \{x_{\boldsymbol{u}} : \boldsymbol{u} \in \{0,1\}^\ell\} = \{x_\emptyset\} \cup \{x_0, x_1\} \cup \{x_{00}, x_{01}, x_{10}, x_{11}\} \cup \ldots$$

*such that for every path $\boldsymbol{y} \in \{0,1\}^d$ and every finite $n < d$, there exists $h \in \mathbb{H}$ so that $h(x_{\boldsymbol{y}_{\leq \ell}}) = y_{\ell+1}$ for $0 \leq \ell \leq n$. We say that $\mathbb{H}$ has an infinite Littlestone tree if there is a Littlestone tree for $\mathbb{H}$ of depth $d = \infty$.*

**Definition 6** (VCL Tree [BHM$^+$21]). *A Vapnik-Chervonenkis-Littlestone (VCL) tree for $\mathbb{H} \subseteq \{0,1\}^{\mathcal{X}}$ of depth $d \leq \infty$ consists of a collection*

$$\bigcup_{0 \leq \ell < d} \{x_{\boldsymbol{u}} \in \mathcal{X}^{\ell+1}, \boldsymbol{u} \in \{0,1\} \times \{0,1\}^2 \times \ldots \times \{0,1\}^\ell\}$$

*such that for every finite level $n < d$, and $\boldsymbol{y} \in \{0,1\} \times \{0,1\}^2 \times \{0,1\}^{n+1}$, there exists some $h \in \mathbb{H}$ such that $h(x^i_{\boldsymbol{y}_{\leq k}}) = y^i_{k+1}$ for all $0 \leq i \leq k$ and $0 \leq k \leq n$, where we denote*

$$\boldsymbol{y}_{\leq k} = (y^0_1, (y^0_2, y^1_2), \ldots, (y^0_k, \ldots, y^{k-1}_k)), \quad x_{\boldsymbol{y}_{\leq k}} = (x^0_{\boldsymbol{y}_{\leq k}}, \ldots, x^k_{\boldsymbol{y}_{\leq k}}).$$

*We say that $\mathbb{H}$ has an infinite VCL tree if it has a VCL tree of depth $d = \infty$.*

**Definition 7** (VC Dimension of Partial Concept Classes [AHHM21])**.** *For a partial concept class $\mathcal{F} \subseteq \{0, 1, \star\}^{\mathcal{X}}$, the VC dimension of $\mathcal{F}$ is defined to be the largest number $d \in \mathbb{N}$ such that $\exists (x_1, \ldots, x_d) \in \mathcal{X}^d$ such that $\{(f(x_1), \ldots, f(x_d)) : f \in \mathcal{F}\} = \{0,1\}^d$. Such a sequence $(x_1, \ldots, x_d)$ is said to be shattered by $\mathcal{F}$. If there is no bound on $d$ we say that the VC dimension is $\infty$.*

## A.3 Omitted Preliminaries

**Gale-Stewart Games.** We briefly discuss some important facts about Gale-Stewart games. Our discussion follows [BHM+21]. We refer to their work for further details and pointers. Let us fix sequences of sets $\mathcal{X}_t, \mathcal{Y}_t$ for $t \geq 1$. We consider infinite games between two players where in each round $t \geq 1$, the first player $P_A$ selects an element $x_t \in \mathcal{X}_t$, and then player $P_L$ selects an element $y_t \in \mathcal{Y}_t$. The rules of the game are determined by defining a set $\mathcal{W} \subseteq \prod_{t \geq 1}(\mathcal{X}_t \times \mathcal{Y}_t)$ of winning sequences for $P_L$. This means that after an infinite sequence of consecutive plays $x_1, y_1, x_2, y_2, \ldots$, we say that $P_L$ wins if $(x_1, y_1, x_2, y_2, \ldots) \in \mathcal{W}$; otherwise the winner is $P_A$.

A *strategy* is a rule used by a given player to determine the next move given the current state and the history of the game. A strategy for $P_A$ is a sequence of functions $f_t : \prod_{s < t}(\mathcal{X}_s \times \mathcal{Y}_s) \to \mathcal{X}_t$ for $t \geq 1$, so that $P_A$ plays $x_t = f_t(x_1, y_1, \ldots, x_{t-1}, y_{t-1})$ in round $t$. Similarly, a strategy for $P_L$ is a sequence of $g_t : \prod_{s<t}(\mathcal{X}_s \times \mathcal{Y}_s) \times \mathcal{X}_t \to \mathcal{Y}_t$ for $t \geq 1$, so that $P_L$ plays $y_t = g_t(x_1, y_1, \ldots, x_{t-1}, y_{t-1}, x_t)$ in round $t$. A strategy for $P_A$ is called *winning* if playing that strategy always makes $P_A$ win the game no matter what $P_L$ plays; a winning strategy for $P_L$ is defined similarly. The prominent question in these infinite games is to come up with conditions under which one of the two players has a winning strategy in the game. Such a condition was introduced by [GS53]: a $\mathcal{W}$ is *finitely decidable* if for every sequence of plays $(x_1, y_1, x_2, y_2, \ldots) \in \mathcal{W}$, there exists some $n < \infty$ so that

$$(x_1, y_1, \ldots, x_n, y_n, x'_{n+1}, y'_{n+1}, x'_{n+2}, y'_{n+2}, \ldots) \in \mathcal{W}$$

for all choices of $x'_{n+1}, y'_{n+1}, x'_{n+2}, y'_{n+2}, \ldots$ In words, the condition that "$\mathcal{W}$ is finitely decidable" means that if $P_L$ wins, then she knows that she won after playing a finite number of rounds. Conversely, $P_A$ wins the game when $P_L$ does not win after any finite number of rounds.

An infinite game whose set $\mathcal{W}$ is finitely decidable is called a *Gale-Stewart game*. An important result for Gale-Stewart games follows.

**Remark 3.** *[GS53, HW+93, Kec12] In any Gale-Stewart game, either $P_A$ or $P_L$ has a winning strategy.*

The above existential result provides no information, however, about the complexity of the winning strategies. Importantly, it is unclear whether winning strategies can be chosen to be measurable. The next result addresses this concern.

**Theorem 4** (Theorem B.1 from [BHM+21])**.** *Let $\{X_t\}_{t \geq 1}$ be Polish spaces and $\{Y_t\}_{t \geq 1}$ be countable sets. Consider a Gale-Stewart game whose set $\mathcal{W} \subseteq \prod_{t \geq 1}(X_t \times Y_t)$ of winning strategies for $P_L$ is finitely decidable and coanalytic. Then there is a universally measurable winning strategy.*

The following remark shows that we can, equivalently, let the strategy of the the learner $P_L$ and the adversary $P_A$ depend only on the choices of their opponent in the previous rounds.

**Remark 4.** *[BHM+21] The strategy of $P_L$ is defined to be a sequence of functions $y_t = f_t(x_1, y_1, \ldots, x_{t-1}, y_{t-1})$ of the history of the game, where $y_1, \ldots, y_{t-1}$ are defined similarly. Thus, we can equivalently let $y_t = f_t(x_1, \ldots, x_{t-1})$. The same holds for the strategy of $P_A$.*

**Partial Concept Classes.** The traditional PAC learning framework handles *total* concept classes, i.e., classes of functions $h : \mathcal{X} \to \{0, 1\}$ that are defined on every point $x \in \mathcal{X}$. The caveat with total functions is that they do not provide a direct way to express *data-dependent* assumptions. For example, the space $\mathcal{X}$ might be high-dimensional but the data that the leaner has to classify could lie

in a low-dimensional space. A prominent application in which this holds is classification of images of animals; the space $\mathcal{X}$ is the set of all possible values of the pixels of the image but most of these configurations of the pixels do not even correspond to an image of an animal. [AHHM21] proposed an extension of the PAC framework that allows one to capture such assumptions using *partial* concept classes, i.e., sets of functions $f : \mathcal{X} \to \{0, 1, \star\}$, where $f(x) = \star$ means that $f$ is undefined at $x$. A lot of notions, such as PAC learnability and the VC dimension, are extended naturally from the setting of total concept classes to the setting of partial concept classes (see, e.g., Definition 7). To illustrate how one can use partial classes, to express data-dependent assumptions, we comment that the class of $d$-dimensional linear classifiers with margin $\gamma > 0$ can be formulated as a partial class: we say that a sample $(x_1, y_1), \ldots, (x_n, y_n) \in \mathbb{R}^d \times \{0, 1\}$ is $(R, \gamma)$-separable if all the points $x_1, \ldots, x_n$ lie in a (euclidean) ball of radius $R$, the 0-labeled examples and the 1-labeled examples are linearly separable, and the (euclidean) distance between the 0-labeled examples and 1-labeled examples is at least $2\gamma$. Then, the class

$$\mathcal{F}_{R,\gamma} = \big\{ f : \mathbb{R}^d \to \{0, 1\star\} : (\forall x_1, \ldots, x_n) \in \mathrm{supp}(f) :$$
$$(x_1, (f(x_1)), \ldots, (x_n, (f(x_n)) \text{ is } (R, \gamma) - \text{separable} \big\},$$

where $\mathrm{supp}(f)$ is the set of all points where $f(x) \neq \star$, expresses the set of functions that satisfy these constraints [AHHM21]. Remarkably, the VC dimension of $\mathcal{F}$ is bounded by $O(\frac{R^2}{\gamma^2})$ [AHHM21]. As another example, we can formulate the constraint that the data have to lie in a low-dimensional space by defining the partial concept class

$$\mathcal{F} = \big\{ f : \mathbb{R}^d \to \{0, 1, \star\} : \dim(\mathrm{supp}(f)) \ll d \big\},$$

where $\dim(S)$ captures the dimension of the set of points in $S$. Interestingly, even though the PAC learnability of partial classes is characterized by the VC dimension (as it is the case with total classes), the ERM algorithm provably fails to learn partial classes. Thus, the algorithmic landscape is much richer compared to total classes. Moreover, there is no systematic way to extend a partial concept class to a total concept class without significantly increasing its VC dimension. For details, we refer to [AHHM21].

**One-Inclusion Graph Algorithm.** We state formally the guarantees of the one-inclusion graph algorithm [HLW94].

**Theorem 5** (One-Inclusion Graph Algorithm [HLW94])**.** *For any (total) concept class $\mathbb{H}$ whose VC dimension is bounded by $\mathrm{d} < \infty$, there is an algorithm $\mathbb{A} : (\mathcal{X} \times \{0, 1\})^* \times \mathcal{X} \to \{0, 1\}$ such that for any $n \in \mathbb{N}$ and any sequence $\{(x_1, y_1), \ldots, (x_n, y_n)\} \in (\mathcal{X} \times \{0, 1\})^n$ that is realizable w.r.t. $\mathbb{H}$,*

$$\frac{1}{n!} \sum_{\sigma \in \mathrm{Sym}(n)} \mathbb{1}\{\mathbb{A}(x_{\sigma(1)}, y_{\sigma(1)}, \ldots, x_{\sigma(n-1)}, y_{\sigma(n-1)}, x_{\sigma(n)}) \neq y_{\sigma(n)}\} \leq \frac{\mathrm{d}}{n},$$

*where $\mathrm{Sym}(n)$ denotes the symmetric group of permutations of $\{1, \ldots, n\}$.*

In particular, Theorem 5 implies immediately that if $(x_1, y_1), \ldots, (x_{n+1}, y_{n+1})$ are i.i.d. from P then the classifier $\tilde{h}_n(\cdot) := \mathbb{A}(x_1, y_1, \ldots, x_n, y_n, \cdot)$ has $\mathbb{E}[\mathrm{er}(\tilde{h}_n)] \leq \frac{\mathrm{d}}{n+1}$.

We remark that, as shown in [AHHM21], this results also holds for partial concept classes.

# B   Arbitrarily Fast Rates

We start with the proof of the first case of the trichotomy we have stated, i.e. that $\mathbb{H}$ is learnable at an arbitrarily fast rate if and only if it does not have an infinite Littlestone tree. Our proof consists of two parts. Firstly, we show that if $\mathbb{H}$ does not have an infinite Littlestone tree, then it is learnable at an arbitrarily fast rate. Then, we show that whenever $\mathbb{H}$ has an infinite Littlestone tree, the best rate we can hope for is exponential.

**Online Adversarial Learning.** We briefly describe the classical adversarial online learning setting. It consists of a game that is played between two players, the adversary $\mathrm{P_A}$ and the learner $\mathrm{P_L}$ in a sequence of discrete rounds, where in every round $t \in \mathbb{N}$ player $\mathrm{P_A}$ presents an instance $x_t \in \mathcal{X}$ to $\mathrm{P_L}$ who then has to guess its label. Subsequently, the adversary, who knows the choice of the learner,

reveals the true label $y_t \in \{0,1\}$ of the point $x_t$. The constraint is that after every round $t$ there has to be some $h_t \in \mathbb{H}$ that is consistent with the execution of the game, i.e., $h_t(x_\tau) = y_\tau, 1 \leq \tau \leq t$. An important component of our construction is an algorithm that guarantees a finite number of mistakes for $\mathrm{P_L}$ in the adversarial online game, when $\mathbb{H}$ does not have an infinite Littlestone tree.

**Theorem 6** ([BHM⁺21]). *Assume that $\mathbb{H}$ does not have an infinite Littlestone tree. Then, the Standard Optimal Algorithm (SOA), extended appropriately to handle infinite ordinal Littlestone dimensions, guarantees a finite number of mistakes in the online game.*

Essentially, this algorithm works in the following way: in every round $t$ player $\mathrm{P_L}$ predicts the label $\hat{y}_t \in \{0,1\}$ so that the class

$$\mathbb{H}_{x_1,y_1,\ldots,x_{t-1},y_{t-1},x_t,\hat{y}_t} := \{h \in \mathbb{H} : h(x_i) = y_i, 1 \leq i \leq t-1, h(x_t) = \hat{y}_t\},$$

has the largest *ordinal* Littlestone dimension. Roughly speaking, the ordinal Littlestone dimension quantifies "how infinite" the Littlestone dimension of the class of consistent classifiers is. If there is a uniform bound on the depth of the Littlestone tree, then this is just a finite number $d \in \mathbb{N}$, so the notion of the ordinal Littlestone dimension coincides with the well-studied notion of Littlestone dimesnsion. The intuition is that either $\mathrm{P_L}$ will predict the correct label and, thus, will not make a mistake, or if she makes a mistake the ordinal Littlestone dimension of the class of consistent hypotheses will decrease. Importantly, [BHM⁺21] showed that if $\mathbb{H}$ does not an infinite Littlestone tree then the ordinal Littlestone dimension does not admit decreasing chains of infinite length. Thus, this strategy of the learner guarantees a finite number of mistakes.

The algorithm that achieves exponentially fast rates, given that $\mathbb{H}$ does not have an infinite Littlestone tree, is outlined in Figure 2.

**Theorem 7.** *If $\mathbb{H}$ does not have an infinite Littlestone tree, then it is interactively learnable with arbitrarily fast rates.*

*Proof.* From the stream of points $\{x_1, x_2, \ldots, \}$ the learner has access to, we define two sets $S_1 = \{x_1, \ldots, x_{m_1}\}$ of $m_1$ points, and $S_2 = \{x_{m_1+1}, \ldots, x_{m_1+m_2}\}$ of $m_2$ points. We split up the first set into $\lfloor \sqrt{m_1} \rfloor$ different batches where the $i^{\text{th}}$ batch is $b_i = \{x_{(i-1)\lfloor \sqrt{m_1} \rfloor+1}, \ldots, x_{i\lfloor \sqrt{m_1} \rfloor}\}$. For every batch $b_i$, we consider the set of all its labeled prefixes in lexicographical order, i.e., $(x_{(i-1)\lfloor \sqrt{m_1} \rfloor+1}, 0), (x_{(i-1)\lfloor \sqrt{m_1} \rfloor+1}, 1), (x_{(i-1)\lfloor \sqrt{m_1} \rfloor+1}, 0, x_{(i-1)\lfloor \sqrt{m_1} \rfloor+2}, 0), \ldots$, and denote by $b_{i,j}, 1 \leq j \leq 2^{\lfloor \sqrt{m_1} \rfloor+1}$, the $j^{\text{th}}$ labeled prefix of the batch $b_i$. Suppose that $\mathbb{H}$ does not have an infinite Littlestone tree. Let $\mathbb{A}_0$ be the ordinal SOA described in Theorem 6 and denote by $\mathbb{A}_0(x; b_{i,j})$ the output of the learner on point $x$ assuming $b_{i,j}$ is the (labeled) set it has been trained on. Consider the set of experts based on $\mathbb{A}_0$: that is, for every labeled batch $b_{i,j}$, there is $g_{i,j} : \mathcal{X} \to \{0,1\}$, such that $g_{i,j}(x) := \mathbb{A}_0(x; b_{i,j})$, i.e., the output of the learner that is trained on $b_{i,j}$. Ideally, we would like to split all the $g_{i,j}$ into equivalence classes, so that each class consists of functions that have the same output on $\mathcal{X}$ (except maybe for a measure-zero set). Since we do not have access to $\mathrm{P}_\mathcal{X}$, we execute this step approximately and we estimate the sets by evaluating the functions $g_{i,j}$ on the points in the set $S_2$. We define a set $\mathbb{F}$ of equivalence classes in the following way: $g_{i,j}$ and $g_{i',j'}$ are in the same class iff they classify $S_2$ the same. If we take $m_2$ to be large enough, since the data in $S_2$ are i.i.d. from $\mathrm{P}_\mathcal{X}$, we can guarantee that $\forall \delta_1, \varepsilon_1 > 0$ with probability at least $1 - \delta_1$, for every $F \in \mathbb{F}$ we have that $\max g_{i,j}, g_{i',j'} \in F \mathrm{P}_\mathcal{X}(x : g_{i,j}(x) \neq g_{i',j'}(x)) \leq \varepsilon_1$. In particular, the probability that a fixed pair $g_{i,j}, g_{i',j'}$ with $\mathrm{P}_\mathcal{X}(x : g_{i,j}(x) \neq g_{i',j'}(x)) \geq \varepsilon_1$ falls into the same equivalence class is bounded by $(1-\varepsilon_1)^{m_2} \leq e^{-\varepsilon_1 m_2}$, so by taking a union bound over all the pairs of classifiers we see that $m_2 \geq \frac{\ln(m_1 2^{2\lfloor \sqrt{m_1} \rfloor+2}/\delta_1)}{\varepsilon_1}$ suffices. The parameters $\varepsilon_1, \delta_1$ will be defined later. Let $\mathcal{E}_1$ be this event. We condition on $\mathcal{E}_1$ for the rest of the proof. Moreover, if $g_{i,j} \in F, g_{i',j'} \notin F$, then $\mathrm{P}_\mathcal{X}(x : g_{i,j}(x) \neq g_{i',j'}(x)) > 0$, with probability one. For each $F \in \mathbb{F}$, we define

$$\mathrm{rank}(F) = \min \left\{ r : \exists\, 1 \leq i_1 < i_2 \ldots < i_{\lfloor \sqrt{m_1} \rfloor/3} \leq \lfloor \sqrt{m_1} \rfloor, k_1, \ldots, k_{\lfloor \sqrt{m_1} \rfloor/3} \in [2^{r+1}] \right.$$

$$\left. \text{such that } g_{i_j,k_j} \in F, 1 \leq j \leq \lfloor \sqrt{m_1} \rfloor/3 \right\},$$

where we let $\min\{\emptyset\} = \infty$[3]. Notice that there are at most $3 \cdot 2^{r+1}$ different classes that have rank $r$ or smaller. We now consider an enumeration of the equivalence classes $\mathbb{F} = \{\mathcal{F}_1, \mathcal{F}_2, \ldots\}$ so

---

[3]For simplicity, we assume $\lfloor m_1 \rfloor$ is a multiple of 3.

that $\mathrm{rank}(\mathcal{F}_\ell)$ is non-decreasing in $\ell$, and we pick any $f_\ell \in \mathcal{F}_\ell$ for each $\ell$. Let $f_1, f_2, \ldots,$ be these functions. We now define an active learning algorithm. We are given a budget $n$ on the number of label queries. For each pair $i, j \leq \lfloor\sqrt{n}\rfloor, i \neq j$, we query the label of some $x_s \in S_2$ for which $f_i(x_s) \neq f_j(x_s)$. Notice that, by definition, such an element always exists since $f_i, f_j$ are in different equivalence classes. Hence, there is at most one $f_{\hat{i}}, 1 \leq \hat{i} \leq \lfloor\sqrt{n}\rfloor$, that classifies correctly all the points whose labels we queried. If such a classifier exists, we output $f_{\hat{i}}$, otherwise we output any $f_\ell$. To complete the proof, we need to show that there exists some $i^* \in \mathbb{N}$ with $\mathrm{er}(f_{i^*}) \leq \varepsilon_1$ and $f_{i^*}$ is correct on all the queries, and hence for any $n > (i^*)^2$ we will output $\hat{h}_n = f_{i^*}$. As in [BHM$^+$21], we know there is a finite distribution-dependent number $m^*$ such that, running $\mathbb{A}_0$ sequentially through $m^*$ i.i.d. labeled points $S_{m^*} = (x_1, y_1, \ldots, x_{m^*}, y_{m^*})$, we have that

$$\Pr(\mathrm{er}(\mathbb{A}_0(\cdot; S_{m^*})) = 0) \geq \frac{1}{2},$$

i.e., $m^*$ is the median time of the last mistake by $\mathbb{A}_0$. Assume that $\lfloor\sqrt{m_1}\rfloor > m^*$ and let $F^*$ be the equivalence class that contains the classifiers that have zero error. We will show that such a class exists, with high probability, and has $\mathrm{rank}(F^*) \leq m^*$. Since we are considering all the possible labelings of i.i.d. data from $P_\mathcal{X}$, for every batch $b_i$ there is some $j_{i^*} \in [2^{m^*+1}]$ such that

$$\Pr\left(\mathrm{er}\left(g_{i,j_i^*}\right) = 0\right) \geq \frac{1}{2}.$$

Thus, using Hoeffding's inequality we see that with probability $1 - e^{-\lfloor\sqrt{m_1}\rfloor/18}$ at least $\lfloor\sqrt{m_1}\rfloor/3$ of the classifiers $g_{i,j}$ will have zero error, where $i \in [\lfloor\sqrt{m_1}\rfloor], j \in [2^{m^*+1}]$. Let $\mathcal{E}_2$ be this event. We condition on this event for the remaining of the proof. Now notice that with probability one, when evaluated on $S_2$, all these zero-error classifiers will produce the same output so they will be in the same class $F^*$. Moreover, since $j \leq 2^{m^*+1}$ we have that $\mathrm{rank}(F^*) \leq m^*$, so its position in the enumeration will be at most $i^* \leq 3 \cdot 2^{m^*+1}$. Now since $F^*$ contains a zero-error classifier we have that the error rate of its representative is $\mathrm{er}(f_{i^*}) \leq \varepsilon_1$, under the event $\mathcal{E}_1 \cap \mathcal{E}_2$. Putting everything together, we see that if $n > \left(3 \cdot 2^{m^*+1}\right)^2$ then

$$\mathbb{E}\left[\mathrm{er}(\hat{h}_n)\right] \leq \varepsilon_1 + \delta_1 + e^{-\lfloor\sqrt{m_1}\rfloor/18}.$$

In order to achieve learning rate $R(n)$, we need to pick $m_1 > \lceil\ln(R(n)^{-1})^2\rceil$ and $\varepsilon_1, \delta_1 = R(n)$. Moreover, notice that since $m_1$ is increasing in $n$, after some finite $n^* \in \mathbb{N}$ we have that $\lfloor\sqrt{m_1}\rfloor > m^*$. $\qquad\square$

We remark that if the learner has access to $P_\mathcal{X}$ all the steps in the proof in which we used data to approximate the probability of some events can be executed exactly, so we get expected error rate that is zero after a finite $n \in \mathbb{N}$.

## B.1 Slower than arbitrarily fast is not faster than exponential

We now provide the lower bound that completes our characterization for the first case of the trichotomy. More precisely, we show that if the class $\mathbb{H}$ has an infinite Littlestone tree, then no algorithm can achieve a learning rate that is faster than exponential. Our approach leverages an impossibility result from rate distortion theory. We first provide some definitions related to rate distortion theory.

**Definition 8.** *Let $\mathcal{D}^*$ be an arbitrary collection of objects and $\mathcal{D} \subseteq \mathcal{D}^*$ We define a* code *to be a pair of measurable functions $(C, D)$. The* encoder *$C$ maps any element $x \in \mathcal{D}$ to a binary sequence $C(x) \in \bigcup_{q=0}^{\infty}\{0, 1\}^q$. The* decoder *$D$ maps any element $c \in \bigcup_{q=0}^{\infty}\{0, 1\}^q$ to an element $D(c) \in \mathcal{D}^*$.*

**Definition 9.** *Let $(C, D)$ be a code. For any $q \in \{0, 1, \ldots, \}, C(x) \in \{0, 1\}^q$ we let $|C(x)| = q$ be the* length *of $C(x)$ and $C^i(x)$ be the $i-$th bit of the codeword. The code is called* prefix-free *if every $x_1, x_2 \in \mathcal{D}$ with $x_1 \neq x_2$ and $|C(x_1)| \leq |C(x_2)|$ have $C^i(x_1) \neq C^i(x_2)$ for some $i : 1 \leq i \leq |C(x_1)|$.*

Let $\rho(\cdot, \cdot)$ be a pseudo-metric on $\mathcal{D}^*$. Let $\pi$ be a probability measure on $\mathcal{D}$ (under the Borel $\sigma$-algebra induced on $\mathcal{D}$ by $\rho$). The *distortion* of code $(C, D)$ is defined as $\mathbb{E}_{x^*\sim\pi}[\rho(x^*, D(C(x^*)))]$, while the *rate* of code $(C, D)$ is defined as $\mathbb{E}_{x^*\sim\pi}[|C(x^*)|]$.

The following result is very important in the derivation of our lower bounds. It effectively represents a universal-rates variant of a type of result common to rate-distortion theory.

**Lemma 1.** *Let $\mathcal{D} = \mathcal{D}^*$ be the set of all countably infinite binary sequences. Let $\pi$ be the uniform distribution over $\mathcal{D}$ (i.e., each bit is independent* Bernoulli(1/2)*). For any $d \in \mathbb{N}$, for $x^*, x \in \{0,1\}^\infty$, let $\rho_d(x^*, x) = \frac{1}{d} \sum_{i=1}^{d} \mathbb{1}[x_i \neq x_i^*]$. Then, for any sequence of prefix-free codes $(C_d, D_d)$ with $|C_d(x^*)| \leq d/32$, with probability one,*

$$\liminf_{d \to \infty} \rho_d(x^*, D_d(C_d(x^*))) \geq 1/4.$$

*Proof.* Let $x^* \sim \pi$, and suppose $(C_d, D_d)$ is a sequence of prefix-free codes with $|C_d(x^*)| \leq d/32$. Let $V_d = \{C_d(x) : x \in \{0,1\}^\infty\}$. Note that, since $(C_d, D_d)$ is prefix-free, $|V_d| \leq 2^{d/32}$. For each $v \in V$, $\mathbb{E}[\rho_d(x^*, v)] = 1/2$, and thus a Chernoff bound implies $\Pr(\rho_d(x^*, v) \leq 1/4) \leq e^{-d/16}$. By the union bound,

$$\Pr(\exists v \in V_d : \rho_d(x^*, v) \leq 1/4) \leq |V_d| e^{-d/16} \leq 2^{d/32} e^{-d/16} \leq e^{-d/32}.$$

Altogether, we have that

$$\Pr(\rho_d(x^*, D_d(C_d(x^*))) \leq 1/4) \leq e^{-d/32}.$$

Since $\sum_{d \in \mathbb{N}} e^{-d/32} < \infty$, the Borel-Cantelli lemma implies that with probability one,

$$\liminf_{d \to \infty} \rho_d(x^*, D_d(C_d(x^*))) \geq \frac{1}{4}.$$

This completes the proof. $\qquad\square$

We now describe the intuition behind our lower bound. We define the target classification using a randomly chosen path in the Littlestone tree (with some care to ensure this is a well-defined classification). This sequence of left/right branches can be represented as an infinite binary string $x^*$. The binary responses to the learner's queries represent a codeword, and the learner's predictions on the points along the target path represent the decoded output of a code. We define the marginal distribution $P_{\mathcal{X}}$ in a way that ensures that having $\rho_d(x^*, D_d(C_d(x^*))) \geq \frac{1}{4}$ implies error rate lower bounded by an exponential $e^{-\Omega(d)}$. The result then follows from Lemma 1.

Before presenting the formal details, we first require a technical lemma, guaranteeing that we may assume, without loss of generality, that the nodes of the infinite Littlestone tree are all distinct.

**Lemma 2.** *If $\mathbb{H}$ has an infinite Littlestone tree, then it also has an infinite Littlestone tree $\{x_{\boldsymbol{u}} : 0 \leq k < \infty, \boldsymbol{u} \in \{0,1\}^k\}$ (in the notation of Definition 5) such that every $x_{\boldsymbol{u}}$ and $x_{\boldsymbol{u}'}$ with $\boldsymbol{u} \neq \boldsymbol{u}'$ have $x_{\boldsymbol{u}} \neq x_{\boldsymbol{u}'}$: that is, all nodes in the tree are distinct.*

*Proof.* Consider any infinite Littlestone tree for $\mathbb{H}$. We will construct an infinite Littlestone tree for which all nodes are distinct by modifying the tree in a breadth-first way. We keep the root node as is. Then, for the purpose of an inductive construction, suppose we have already ensured that the first $n-1$ nodes in the breadth-first order are distinct – these points will remain unchanged forevermore in this modification process – and that otherwise the rest of the tree remains a valid infinite Littlestone tree for $\mathbb{H}$. Next we wish to specify the $n^{\text{th}}$ node in this breadth-first order, and modify the tree to remain a valid infinite Littlestone tree. For the subtree rooted at the node that is currently in the $n^{\text{th}}$ position (in the current construction-so-far), since every node along any given infinite path of descendants of this node must be distinct (otherwise branching in opposite ways for the two nodes would not be realizable), there are necessarily an infinite number of distinct points $x$ to be found within its subtree. Since there are only $n-1$ nodes that are "fixed" so far, there must exist points $x$ in this subtree that are distinct from all $n-1$ previously chosen points. Choose some such $x$ and let $T_x$ be the subsubtree rooted at $x$ in the current tree. Now define a modified tree, in which the $n^{\text{th}}$ node in the breadth-first order is replaced by this $x$, and the subtree rooted at this node is $T_x$; everything else remains unchanged. By construction, the first $n$ nodes in the breadth-first order are now distinct. Moreover, every finite-depth path in the modified tree which does not pass through this $n^{\text{th}}$ node is unchanged, and hence remains realizable. On the other hand, since the ancestors and descendants of this new $n^{\text{th}}$ node in the modified tree were already ancestors and descendants, respectively, in the tree before this modification, every finite-depth path in the modified tree which passes through this $n^{\text{th}}$ node is a subset of a finite-depth path in the tree from before this modification; since the tree before the modification was a valid infinite Littlestone tree for $\mathbb{H}$, it must be that this

path was realizable, and therefore so is its sub-path that remains in the tree after the modification. Thus, the tree remains a valid infinite Littlestone tree for $\mathbb{H}$. Continuing this construction inductively, we arrive at an infinite Littlestone tree such that all nodes are distinct (since otherwise, if two nodes were identical, one would precede the other in the breadth-first order, contradicting the invariant maintained by the induction). $\qquad\square$

We are now ready for the proof that classes with an infinite Littlestone tree are not interactively learnable at rates faster that $e^{-n}$.

**Theorem 8.** *If $\mathbb{H}$ has an infinite Littlestone tree, then $\mathbb{H}$ is not interactively learnable at rate faster than exponential: $e^{-n}$. This holds even if $\mathrm{P}_{\mathcal{X}}$ is known to the learner.*

*Proof.* As mentioned, the idea of the proof is to set up an equivalence to the coding problem stated above in Lemma 1, where the binary responses to the learner's queries are the code words. Suppose $\mathbb{H}$ has an infinite Littlestone tree, and let $T = \{x_{\boldsymbol{u}} : 0 \leq k < \infty, \boldsymbol{u} \in \{0,1\}^k\}$ be any such infinite tree for which the points $x_{\boldsymbol{u}}$ are all distinct; such a tree is guaranteed to exist by Lemma 2. We begin by specifying the marginal probability distribution $\mathrm{P}_{\mathcal{X}}$ on $\mathcal{X}$. For the points $x_{\boldsymbol{u}}$ with $\boldsymbol{u} \in \{0,1\}^{i-1}$ (i.e., nodes at depth $i$ in the tree), we set $\mathrm{P}_{\mathcal{X}}(\{x_{\boldsymbol{u}}\}) = 2^{1-2i}$. Since every $x_{\boldsymbol{u}}$ is distinct, this value is well-defined. Note that, since there are $2^{i-1}$ nodes at depth $i$ (counting the root to be depth 1), the total probability mass at depth $i$ is $2^{-i}$, and hence $\mathrm{P}_{\mathcal{X}}(T) = 1$, so that this completes the definition of $\mathrm{P}_{\mathcal{X}}$.

Next, we define the target labeling of the points via the probabilistic method. Let $\boldsymbol{y} = (y_1, y_2, \ldots,)$ be a sequence of i.i.d. Bernoulli$(1/2)$ random variables. We consider the random path of the tree that is induced by $\boldsymbol{y}$. Let $x_1, x_2, \ldots,$ be the elements that are on this path: that is, $x_i = x_{(y_1,\ldots,y_{i-1})}$ in the notation of the tree $T$. The target labels $h^*(x_i)$ of these points are determined by $\boldsymbol{y}$: $h^*(x_i) = y_i$.

It remains to specify the target labels $h^*(x)$ for nodes $x \in T$ that are not among $\{x_1, x_2, \ldots\}$. We will specify their labels in a breadth-first manner, as follows. For each $i \in \mathbb{N}$, let $h_i$ be a classifier in $\mathbb{H}$ that realizes these labels up to depth $i$: i.e., $h_i(x_j) = y_j$ for all $j \leq i$. Such an $h_i$ must exist in $\mathbb{H}$ for all $i \in \mathbb{N}$, since all finite-depth paths are realizable by $\mathbb{H}$. We define $V = \{h_i : i \in \mathbb{N}\}$. To specify the target labels for the remaining points, we inductively follow a breadth-first traversal. Consider the next point $x$ in this traversal. To determine the label we assign for $h^*(x)$, if there are infinitely many $h \in V$ with $h(x) = 0$, we set the label $h^*(x) = 0$, and otherwise we set its label as $h^*(x) = 1$. Either way, we update $V$ by discarding the classifiers that disagree with the label we assign for $x$. We then continue on to the next $x$ in the breadth-first traversal. Notice that by construction, $V$ initially contains an infinite number of functions, so at every $x$, either infinitely many $h \in V$ have $h(x) = 0$ or infinitely many $h \in V$ have $h(x) = 1$; either way, we assign a label that maintains the invariant that $V$ remains infinite after pruning the inconsistent functions. By induction, this specifies a target classification $h^*(x)$ for every point in $T$. Moreover, since $T$ is the (countable) support of $\mathrm{P}_{\mathcal{X}}$, and since $V$ is always non-empty, even after constraining to agree with $h^*$ labels on any finite number of points in the breadth-first order, we find that $\mathrm{P}_{\mathcal{X}}$ and $h^*$ together specify a realizable distribution on $\mathcal{X} \times \{0,1\}$. Moreover, by the definition of queries in this work, we know there exists a $\theta^* \in \Theta^*$ for which $h(\theta^*, \cdot)$ takes the $h^*$ classifications of points in $T$; thus, we may take this $\theta^*$ value to define $h^* = h(\theta^*, \cdot)$, and the learner's queries will receive responses consistent with this $\theta^*$.

Now let $\mathbb{A}^*$ be any learning algorithm First, we will use this algorithm to create a sequence of codes $(C_d, D_d)$ for infinite binary strings, under the pseudo-metrics $\rho_d$ from Lemma 1, where the query budget $n = n_d := \lceil d/32 \rceil - 1$, and $d > 32$. Notice that our choice $\boldsymbol{y}$ of the path in the tree is equivalent to choosing a binary string uniformly at random. Moreover, conditioned on the unlabeled data, $\boldsymbol{y}$ completely determines the answers to any queries the learner could ask (since it determines $h^*$, and $\mathrm{P}_{\mathcal{X}}$ is considered fixed). For a budget $n_d$ on the number of queries, let $C_d(\boldsymbol{y})$ be the string of binary responses to the algorithm's queries; in particular, $|C_d(\boldsymbol{y})| \leq n_d < d/32$. If we condition on the unlabeled data and any internal randomness of $\mathbb{A}^*$, then $C_d(\boldsymbol{y})$ will be purely determined by $\boldsymbol{y}$, and moreover, will be prefix-free, since (conditioned on the unlabeled data and the internal randomness) a decision of whether to stop querying early could also only depend on the answers returned for its queries up til then. Let $\hat{h}_{n_d}$ be the output of $\mathbb{A}^*$ for a budget of $n_d$ queries. Also define a decoder $D_d(C_d(\boldsymbol{y})) = (\hat{h}_{n_d}(x_1), \hat{h}_{n_d}(x_2), \ldots)$.

Let $K$ denote a random variable comprised of the unlabeled data and any internal randomness of the learner $\mathbb{A}^*$. By Lemma 1 (applied under the conditional distribution given $K$) and the law of

total probability, with probability one, $\exists d_0 \in \mathbb{N}$ such that every $d \geq d_0$ has $\rho_d(\boldsymbol{y}, D_d(C_d(\boldsymbol{y}))) \geq \frac{1}{5}$. Again by the law of total probability, this further implies that there exists a choice of $\boldsymbol{y}$ such that, conditioned on $\boldsymbol{y}$, with conditional probability (over $K$) one, $\exists d_0 \in \mathbb{N}$ for which every $d \geq d_0$ has

$$\rho_d(\boldsymbol{y}, D_d(C_d(\boldsymbol{y}))) \geq \frac{1}{5}.$$

In particular, for any such $d$, there must be at least one $i \leq \lfloor (4/5)d \rfloor + 1$ for which $\hat{h}_{n_d}(x_i) \neq y_i$. This implies

$$\text{er}(\hat{h}_{n_d}) \geq 2^{-(8/5)d-1} \geq 2^{-(8/5)32n_d - 256/5 - 1}.$$

Altogether, there exists a deterministic choice $y^*$ of $\boldsymbol{y}$ such that, given $\boldsymbol{y} = y^*$, with probability one, every sufficiently large $d$ satisfies $\text{er}(\hat{h}_{n_d}) \geq e^{-cn_d}$ for a numerical constant $c$. Thus, by Fatou's lemma,

$$\liminf_{d \to \infty} e^{cn_d} \mathbb{E}\left[\text{er}(\hat{h}_{n_d}) \Big| \boldsymbol{y} = y^*\right]$$
$$\geq \mathbb{E}\left[\liminf_{d \to \infty} e^{cn_d}\text{er}(\hat{h}_{n_d}) \Big| \boldsymbol{y} = y^*\right] \geq 1.$$

In other words, the expected error rate is lower bounded by $(1 - o(1))e^{-cn_d}$ for all sufficiently large $d$. In particular, this implies that for this choice of $\boldsymbol{y} = y^*$, $\mathbb{E}[\text{er}(\hat{h}_n)] \geq (1 - o(1))e^{-cn}$ for infinitely many $n$, so that $\mathbb{A}^*$ does not achieve a rate faster than $e^{-n}$. This completes the proof. $\qquad \square$

## C   Exponential Rates

We now prove that if $\mathbb{H}$ does not have an infinite VCL tree, then it admits exponential learning rates. The results presented in Appendix B show that when $\mathbb{H}$ has an infinite Littlestone tree, we cannot achieve rates faster than exponential. Thus, these two results show that the optimal learning rate when $\mathbb{H}$ has an infinite Littlestone tree but does not have an infinite VCL tree is exponential. Finally, we provide a lower bound which shows that if $\mathbb{H}$ has an infinite VCL tree, then it only admits arbitrarily slow rates. This completes our characterization.

### C.1   The VCL game

An important component of our algorithm is a Gale-Stewart game from [BHM+21], called the VCL game. Recall that every node of the VCL tree at depth $n$ consists of $n + 1$ points and there are $2^{n+1}$ edges attached to the node, which are labeled with one of the possible classifications of the points in the node. Intuitively, the VCL game $\mathfrak{G}$ generalizes the classical online adversarial game, in the sense that instead of presenting to the learner one point in every round, the adversary selects a number of points that increases linearly with the number of rounds. In every round $\tau$ we have the following interaction between the learner $P_L$ and the adversary $P_A$:

- Player $P_A$ chooses points $\xi_\tau = (\xi_\tau^0, \ldots, \xi_\tau^{\tau-1}) \in \mathcal{X}^\tau$.
- Player $P_L$ chooses labels $\eta_\tau = (\eta_\tau^0, \ldots, \eta_\tau^{\tau-1}) \in \{0, 1\}^\tau$.
- Player $P_L$ wins the game in round $\tau$ if

$$\mathbb{H}_{\xi_1, \eta_1, \ldots, \xi_\tau, \eta_\tau} := \{h \in \mathbb{H} : h(\xi_s^i) = \eta_s^i, 0 \leq i < s, 1 \leq s \leq \tau\} = \emptyset.$$

If the game does not terminate, then player $P_A$ wins. It is clear that $\mathfrak{G}$ is a Gale-Stewart game since the winning set of $P_L$ is finitely decidable. We use the following result from [BHM+21].

**Lemma 3.** *[BHM+21] If $\mathbb{H}$ does not have an infinite VCL tree, then $P_L$ has a universally measurable winning strategy in the game $\mathfrak{G}$.*

The importance of this game is that it gives rise to a universally measurable *pattern avoidance function*. Essentially, this function takes as input an unlabeled sequence of data points and returns a binary string that *does not* correspond to a valid classification of these points. This is achieved using the algorithm in Figure C.1. Following the notation in [BHM+21], we denote by $\eta_\tau : \prod_{\sigma=1}^\tau \mathcal{X}^\sigma \to \{0, 1\}^\tau$ the

winning strategy of the learner[4]. We let $N = \lfloor n/2 \rfloor$ and we use $N$ queries to find the labels of the points $x_1, \ldots, x_N$ in order to simulate the VCL game.

---

**VCLGameSubroutine** [BHM$^+$21]: Input is a labeled data sequence $\{(x_1, y_1), \ldots, (x_N, y_N)\}$.

    1. Let $\tau_0 \leftarrow 0$.

    2. In every step $t \geq 1$ :
- If $\eta_{t_{\tau-1}}(\xi_1, \ldots, \xi_{\tau_{t-1}-1}, x_{t-\tau_{t-1}+1}, \ldots, x_t) = (y_{t-\tau_{t-1}+1}, \ldots, y_t)$ :
  - Let $\xi_{\tau_{t-1}} \leftarrow (x_{t-\tau_{t-1}+1}, \ldots, x_t)$ and $\tau_t \leftarrow \tau_{t-1} + 1$.
- Else, $\tau_t \leftarrow \tau_{t-1}$.

---

Essentially, this algorithm traverses the data sequence that is given as input and simulates the VCL game that we described before. Every time the learner's prediction is achieved by some subsequence of the data, we proceed to the next round of the simulated game. The idea is that since $\mathbb{H}$ does not have an infinite VCL tree, this game can only proceed a *finite* number of times. Hence, after some finite timestep $t^* \in \mathbb{N}$, the learner's strategy $\eta_{\tau_{t^*}}$ will output *forbidden patterns* in the data, with probability one. Give a tuple $(x_1, \ldots, x_{\tau_{t^*}}) \in \mathcal{X}^{\tau_{t^*}}$ we call a binary pattern $(b_1, \ldots, b_{\tau_{t^*}}) \in \{0, 1\}^{\tau_{t^*}}$ forbidden if it is not the correct labeling of the tuple $(x_1, \ldots, x_{\tau_{t^*}})$. We denote by

$$\hat{\boldsymbol{y}}_{t-1}(z_1, \ldots, z_{\tau_{t-1}}) := \eta_{\tau_{t-1}}(\xi_1, \ldots, \xi_{\tau_{t-1}-1}, z_1, \ldots, z_{\tau_{t-1}})$$

the (universally measurable) function that is induced by Algorithm C.1. To understand the intuition behind our approach, let us first assume we know $t^*$. Then, for any labeled tuple of size $\hat{N} = \tau_{t^*}$, the function $\hat{\boldsymbol{y}}_{t^*}$ produces a forbidden labeling. Since we have pinpointed a constraint that the data need to satisfy, a natural thing to do is to express it via a *partial concept class.* We let

$$\hat{\mathcal{F}} = \left\{ f : \mathcal{X} \to \{0, 1, \star\} : \forall (x_1, \ldots, x_{\hat{N}}) \in \mathcal{X}^{\hat{N}}, f(x_1, \ldots, x_{\hat{N}}) \neq \hat{\boldsymbol{y}}_{t^*}(x_1, \ldots, x_{\hat{N}}) \right\}$$

be the partial concept class that does not produce the forbidden labelings. Notice that, by definition, the VC dimension of $\hat{\mathcal{F}}$ is bounded by $\hat{N}$. Thus, assuming access to $t^*$ (which we will estimate later), we have reduced our problem to interactively learning a *partial* concept class whose VC dimension is bounded. In the next section we provide an algorithm for this task.

## C.2 Interactive learning of partial concept classes with bounded VC dimension

The main result of this section is an algorithm which given access to a partial concept class $\mathcal{F}$ whose VC dimension is bounded by d, achieves exponential learning rates in the interactive learning setting. As in the rest of this work, we assume that P is realizable with respect $\mathcal{F}$, i.e. $\inf_{f \in \mathcal{F}} \mathrm{er}(f) = 0$. In particular, this means that for every *finite* sample $S$ drawn i.i.d. from $P_{\mathcal{X}}$ the set of functions $f \in \mathcal{F}$ that are *not* undefined on $S$ is non-empty, with probability one. This can be seen as follows. Since the distribution P is realizable, there exists a sequence of functions $f_k \in \mathcal{F}$ so that

$$P[f_k(x) \neq y] < \frac{1}{2^k}.$$

Let us fix $m \geq 1$. We have that

$$\sum_{k=1}^{\infty} \mathrm{Pr}[\exists s \leq m : f_k(x_s) \neq y_s] \leq m \sum_{k=1}^{\infty} P[f_k(x) \neq y] < \infty,$$

where the first inequality is due to union bound. By Borel-Cantelli, with probability one, there exists for every $m \geq 1$ a hypothesis $f \in \mathcal{F}$ so that $f(x_s) = y_s$ for all $s \leq m$.

Our result is summarized in the following theorem.

**Theorem 9.** *There exists an interactive learning algorithm* $\mathbb{A}$ *for any partial concept class* $\mathcal{F}$ *whose VC dimension is bounded by* d *that achieves exponential error rate: namely,* $\mathbb{E}[\mathrm{er}(\hat{h}_n)] \leq Cd\mathrm{e}^{-cn/d}$ *for universal constants* $C, c > 0$.

---

[4]We can, equivalently, let the strategy of the learner depend only on the previous choices of the adversary (cf. Remark 4).

We outline the algorithm that achieves this rate in Figure 3.

As we explained, with probability one, it holds that $\hat{\mathbb{H}}|_S \neq \emptyset$. Moreover, the labeled points $(x_1, y_1, \ldots, x_m, y_m)$ are drawn i.i.d. from P. Since the one-inclusion graph algorithm guarantees $\mathbb{E}[\mathrm{er}(\hat{h}_m)] \leq \frac{\mathrm{d}}{m+1}$ (cf. Theorem 5), the correctness of Step 5 implies Theorem 9. The next lemma establishes it.

**Lemma 4.** *For any $m$ unlabeled points from $\mathcal{X}$, and total concept class $\mathbb{H}$ on them with VC dimension at most $\mathrm{d}$, there exists an interactive learning algorithm $\mathbb{A}$ such that, for any unknown true labeling of the $m$ points that is realizable by $\mathbb{H}$, the algorithm identifies all $m$ labels using $O(\mathrm{d} \log m)$ queries.*

*Proof.* Since $\mathbb{H}$ has VC-dimension that is bounded by $\mathrm{d}$, we know that there are at most $N = O(m^{\mathrm{d}})$ possible classifications for these points (this follows from Sauer's lemma [Sau72]). We split these into two disjoint sets $\hat{S}_1, \hat{S}_2$ that have size at most $\lceil N/2 \rceil$. We submit a query asking whether the correct classification is in $\hat{S}_1$. Depending on the answer, we pick the appropriate set and recurse. Hence, after every query the size of the set of the possible classifications decreases by a factor of 2. Thus, we see that $O(\mathrm{d} \log m)$ queries suffice. $\square$

Notice that Lemma 4 shows that if we have a query budget $n$, we can find the labels of $m = O(e^{n/\mathrm{d}})$ points.

The proof of our result follows by feeding these $m$ labeled points to the one-inclusion graph algorithm [HLW94] which outputs a classifier $\hat{h}_m$ with $\mathbb{E}[\hat{h}_m] \leq \frac{\mathrm{d}}{m+1}$ (see Theorem 5). Substituting the value of $m$, we get that $\mathbb{E}[\mathrm{er}(\hat{h}_n)] \leq O(\mathrm{d} e^{-n/\mathrm{d}})$.

## C.3 Exponential learning rates

As we alluded to before, the learner does not have access to the time after which the VCL game has converged. Since the data that are used to simulate the game follow a distribution, the time $t^*$ also follows a distribution that is not known to the learner. Our approach to overcome this obstacle builds upon some techniques developed in [BHM+21]. The idea is that given $N$ labeled points, there is a universally measurable strategy to estimate a batch size $\hat{t}_N$ such that the VCL game executed on a batch of size $\hat{t}_N$ will have converged with probability $5/8$. We first introduce some necessary notation from [BHM+21]. Given some universally measurable pattern avoidance function $g : \mathcal{X}^k \to \{0,1\}^k$, we define its error to be

$$\mathrm{per}(g) = \mathrm{per}^k(g) := \mathrm{P}^{\otimes k}[(x_1, y_1, \ldots, x_k, y_k) : g(x_1, \ldots, x_k) = (y_1, \ldots, y_k)],$$

i.e., this is the probability that $g$ will output a true pattern in the data (which is considered an error since $g$ is intended to specify forbidden patterns). The next lemma from [BHM+21] formalizes the previous discussion.

**Lemma 5.** *[BHM+21]* *For any $N \in \mathbb{N}$, there exists a universally measurable $\hat{t}_N = \hat{t}_n(x_1, y_1, \ldots, x_{\lfloor N/2 \rfloor}, y_{\lfloor N/2 \rfloor})$ whose definition does not depend on P so that the following holds. Let the critical time $t^* \in \mathbb{N}$ (P-dependent) be such that*

$$\mathrm{P}\left[\mathrm{per}(\hat{\boldsymbol{y}}_{t^*}) > 0\right] \leq 1/8 \,.$$

*Then, there exist $C, c > 0$ that depend on $P, t^*$ but not $n$ so that*

$$\mathrm{P}[\hat{t}_N \in T^\star] \geq 1 - Ce^{-cN} \,,$$

*where*

$$T^* = \{1 \leq t \leq t^* : \mathrm{P}\left[\mathrm{per}(\hat{\boldsymbol{y}}_t) > 0\right] \leq 3/8\} \,.$$

We use the first $\lfloor N/2 \rfloor$ points to compute that estimate. Then, we proceed as follows: we split the remaining labeled sequence into $\hat{N} = \lfloor N/2\hat{t}_N \rfloor$ batches of size $\hat{t}_N$ and we know that, with high probability, most of them will produce correct functions $\hat{\boldsymbol{y}}_{\hat{t}_N}^i$ that avoid patterns in the data. However, we do not know which of these functions are correct pattern avoidance functions. Hence, our approach is to *aggregate* them and define a partial concept class $\mathcal{F}_m^{\hat{N}}$ that corresponds to the *majority* of them. To be more precise, for $1 \leq i \leq \hat{N}$, we let

$$\mathcal{F}_i = \left\{f : \mathcal{X} \to \{0, 1, \star\} : \forall(x_1, \ldots, x_{\tau_{\hat{t}_N}^i}) \in \mathcal{X}^{\tau_{\hat{t}_N}^i}, (f(x_1), \ldots, f(x_{\tau_{\hat{t}_N}^i})) \neq \hat{\boldsymbol{y}}_{\hat{t}_n}^i(x_1, \ldots, x_{\tau_{\hat{t}_N}^i})\right\},$$

be the partial concept class that is induced by playing the VCL game on the $i$-th batch of the data. We define the majority class $\mathcal{F}_m^{\hat{N}}$ to be

$$\mathcal{F}_m^{\hat{N}} = \left\{ f : \mathcal{X} \to \{0, 1, \star\} : \forall \ell \in \mathbb{N}, \forall (x_1, \ldots, x_\ell) \in \mathcal{X}^\ell, \exists \text{ at least } (9/16)\hat{N} \text{ classes } \mathcal{F}_j, 1 \le j \le \hat{N} \text{ s.t.} \right.$$

$$\left. \exists \hat{f} \in \mathcal{F}_j \text{ with } (\hat{f}(x_1), \ldots, \hat{f}(x_\ell)) = (f(x_1), \ldots, f(x_\ell)) \right\}.$$

Our goal is to execute the Algorithm 3 on $\hat{F}_m^{\hat{N}}$. Recall that this algorithm requires that the partial concept class has *bounded* VC dimension. The next lemma establishes this result.

**Lemma 6.** *The majority class $\mathcal{F}_m^{\hat{N}}$ has VC dimension that is bounded by some distribution-dependent number $\hat{\mathrm{d}}$, with probability at least $1 - e^{-c\hat{N}}$, for some absolute constant $c > 0$.*

*Proof.* We know that there exists some $\mathrm{d}^*$ such that, with probability at least $9/10$, the VC dimension of any partial concept class $\mathcal{F}_i$ is at most $\mathrm{d}^*$. We let $X_i = \mathbb{1}\{\text{VC dimension of } \mathcal{F}_i > \mathrm{d}^*\}$. Notice the all the $X_i$'s are i.i.d. Bernoulli random variables with $p \le 1/10$, since the data that induce the classes $\mathcal{F}_i$ are i.i.d.. Thus, we can use Hoeffding's inequality to bound the probability that at least more than $2/10$ of them have VC dimension greater than $\mathrm{d}^*$ as follows

$$\Pr\left[ \sum_{i=1}^{\hat{N}} X_i \ge (2/10)\hat{N} \right] = \Pr\left[ \sum_{i=1}^{\hat{N}} X_i - (1/10)\hat{N} \ge (1/10)\hat{N} \right] \le e^{-\hat{N}/50}.$$

We let $\mathcal{E}_1$ be the event above and we condition on it for the rest of the proof. So, we know that at least $8/10\hat{N}$ of the partial concept classes $\mathcal{F}_i$ have VC dimension bounded by $\mathrm{d}^*$. We will bound the size $m$ of the largest set that $\mathcal{F}_m^{\hat{N}}$ shatters. For any sequence $(x_1, \ldots, x_m) \in X^m$ that $\mathcal{F}_m^{\hat{N}}$ shatters, we have that

$$\frac{1}{2^m} \sum_{y \in \{0,1\}^m} \mathbb{1}\left\{ \frac{1}{\hat{N}} \sum_{i=1}^{\hat{N}} \mathbb{1}\{\exists f \in \mathcal{F}_i : (f(x_1), \ldots, f(x_m)) = (y_1, \ldots, y_m)\} > 9/16 \right\} = 1.$$

Using Markov's inequality, we get that

$$\frac{1}{2^m} \sum_{y \in \{0,1\}^m} \frac{1}{\hat{N}} \sum_{i=1}^{\hat{N}} \mathbb{1}\{\exists f \in \mathcal{F}_i : (f(x_1), \ldots, f(x_m)) = (y_1, \ldots, y_m)\} > \frac{9}{16}.$$

Swapping the summation gives us

$$\frac{1}{\hat{N}} \sum_{i=1}^{\hat{N}} \frac{1}{2^m} \sum_{y \in \{0,1\}^m} \mathbb{1}\{\exists f \in \mathcal{F}_i : (f(x_1), \ldots, f(x_m)) = (y_1, \ldots, y_m)\} > \frac{9}{16} \iff$$

$$\frac{1}{\hat{N}} \sum_{i=1}^{\hat{N}} \frac{1}{2^m} \sum_{y \in \{0,1\}^m} \mathbb{1}\{\nexists f \in \mathcal{F}_i : (f(x_1), \ldots, f(x_m)) = (y_1, \ldots, y_m)\} \le \frac{7}{16}.$$

Using Markov's inequality again, we get that

$$\frac{1}{\hat{N}} \sum_{i=1}^{\hat{N}} \mathbb{1}\left\{ \frac{1}{2^m} \sum_{y \in \{0,1\}^m} \mathbb{1}\{\nexists f \in \mathcal{F}_i : (f(x_1), \ldots, f(x_m)) = (y_1, \ldots, y_m)\} > 9/16 \right\} \le$$

$$\frac{16}{9} \cdot \frac{1}{\hat{N}} \sum_{i=1}^{\hat{N}} \frac{1}{2^m} \sum_{y \in \{0,1\}^m} \mathbb{1}\{\nexists f \in \mathcal{F}_i : (f(x_1), \ldots, f(x_m)) = (y_1, \ldots, y_m)\}.$$

This implies that

$$\frac{1}{\hat{N}}\sum_{i=1}^{\hat{N}} \mathbb{1}\left\{\frac{1}{2^m}\sum_{y\in\{0,1\}^m}\mathbb{1}\{\nexists f\in\mathcal{F}_i : (f(x_1),\ldots,f(x_m))=(y_1,\ldots,y_m)\} > 9/16\right\} < 7/9 \iff$$

$$\frac{1}{\hat{N}}\sum_{i=1}^{\hat{N}} \mathbb{1}\left\{\frac{1}{2^m}\sum_{y\in\{0,1\}^m}\mathbb{1}\{\exists f\in\mathcal{F}_i : (f(x_1),\ldots,f(x_m))=(y_1,\ldots,y_m)\} \geq 7/16\right\} \geq 2/9.$$

Thus, at least $2/9$ of the partial concept classes $\mathcal{F}_i$ realize at least $7/16$ of all the possible classifications of $(x_1,\ldots,x_m)$. We know that some of them have VC dimension that is bounded by $d^*$, thus we get that $(em/d^*)^{d^*} \geq 2^{m-2} \implies m = O(d^*)$. Hence, the VC dimension of $\mathcal{F}_m$ is at most $O(d^*)$ and this holds with probability $1 - e^{-c\hat{N}}$. $\square$

Next, we define a sequence of universally measurable functions $\{G_\ell : (\mathcal{X}\times\{0,1\})^\ell \to \{0,1\}\}_{\ell\in\mathbb{N}}$ where $G_\ell(x_1,y_1,\ldots,x_\ell,y_\ell) = \mathbb{1}\left\{\exists f\in\mathcal{F}_m^{\hat{N}} : (f(x_1),\ldots,f(x_\ell))=(y_1,\ldots,y_\ell)\right\}$. An equivalent interpretation of the previous lemma is that the partial class on which $\{G_\ell\}_{\ell\in\mathbb{N}}$ returns 1 has bounded VC dimension, with high probability. The next lemma is a key component in the derivation of our result and it states that $\{G_\ell\}_{\ell\in\mathbb{N}}$ return 1 on all finite subsets of the true labeled data sequence, with high probability.

**Lemma 7.** *The $\{G_\ell\}_{\ell\in\mathbb{N}}$ as defined above are universally measurable functions, and with probability at least $1 - Ce^{-cn}, C,c > 0$, the following hold:*

- *The class*

  $$\mathcal{F}_G = \{f:\mathcal{X}\to\{0,1,*\} : \forall\ell\in\mathbb{N}, \forall(x_1,\ldots,x_\ell)\in\mathcal{X}^\ell : G_\ell(x_1,f(x_1),\ldots,x_\ell,f(x_\ell)) = 1\}$$

  *has VC dimension bounded by some distribution-dependent number $\hat{d}$.*

- $\forall\ell\in\mathbb{N}$, *for* $(x_1,y_1,\ldots,x_\ell,y_\ell)\sim P^\ell$, $G_\ell(x_1,y_1,\ldots,x_\ell,y_\ell) = 1$ *with conditional probability one (given $G_\ell$).*

*Proof.* We condition on the event $\mathcal{E}_0$ described in Lemma 6. Notice that, by definition, the class $\mathcal{F}_G$ shatters a sequence $S = x_1,\ldots,x_\ell \in \mathcal{X}^\ell$ if and only if $\mathcal{F}_m^{\hat{N}}$ shatters $S$. Hence, the bound on the VC dimension follows immediately from Lemma 6. We also condition on the event $\mathcal{E}_1$ described in Lemma 5. We now bound the probability that at least $(13/32)\hat{N}$ of the functions $\hat{y}_{\hat{t}_n}^i$ have positive probability of not avoiding patterns. For any $t \in T^*$, using Hoeffding's inequality we get that

$$\Pr\left[\frac{1}{\hat{N}}\sum_{i=1}^{\hat{N}}\mathbb{1}\{\mathrm{per}(\hat{y}_t^i)>0\} > \frac{13}{32}\right] = \Pr\left[\frac{1}{\hat{N}}\sum_{i=1}^{\hat{N}}\mathbb{1}\{\mathrm{per}(\hat{y}_t^i)>0\} - \frac{3}{8} > \frac{1}{32}\right] \leq \exp\left(-\hat{N}/512\right),$$

Since we know that $\hat{t}_n \leq t^*$, taking a union bound over all $1 \leq t \leq t^*$ we get that

$$\Pr\left[\frac{1}{\hat{N}}\sum_{i=1}^{\hat{N}}\mathbb{1}\{\mathrm{per}(\hat{y}_{\hat{t}_n}^i) > 0\} > \frac{13}{32}, \hat{t}_n \in T^*\right] \leq \sum_{t\in T^*}\Pr\left[\frac{1}{\hat{N}}\sum_{i=1}^{\hat{N}}\mathbb{1}\{\mathrm{per}(\hat{y}_t^i)>0\} > \frac{13}{32}\right]$$

$$\leq t^* \exp\left(-\hat{N}/512\right).$$

Thus, except for an event with exponentially small probability, at least $19/32$ of the functions $\hat{y}_{\hat{t}_n}^i$ have zero error at avoiding patterns in the data, with probability one. Notice that, by definition, if the partial class $\mathcal{F}_i$ that corresponds to such a function cannot produce a labeling $y_\ell \in \{0,1\}^\ell$ of some tuple $x_\ell \in \mathcal{X}^\ell$ where $x_\ell \sim P_X^\ell$, we can infer that this $y_\ell$ is not the correct labeling, and this inference will be valid with probability one over the draw of $x_\ell$. Thus, with probability one, if the $9/16$-majority cannot produce some labeling we have that at least one such partial class $\mathcal{F}_j$ that corresponds to a correct $\hat{y}_{\hat{t}_n}^j$ cannot produce this labeling, so it is not the correct one. As a result, with probability one, if $\mathcal{F}_m^{\hat{N}}$ cannot produce $y_\ell$ for $x_\ell$ we know that this is not the correct labeling.

The measurability of $\{G_\ell\}_{\ell \in \mathbb{N}}$ follows by the measurability of all the $\hat{\boldsymbol{y}}_{\hat{t}_n}^i$. The proof of the lemma follows by noticing that the probability of all the events we have conditioned on can be bounded by $1 - C'e^{-c'\hat{N}}$ and that $\hat{N} = \lfloor n/2\hat{t}_n \rfloor \geq \lfloor n/2t^* \rfloor$. $\qquad\square$

Equipped with the previous result, we are now ready to describe the algorithm that achieves the exponential learning rates in this setting.

**Theorem 10.** *If $\mathbb{H}$ does not have an infinite VCL tree, then $\mathbb{H}$ is learnable at rate $e^{-n}$.*

*Proof.* We consider the execution of Algorithm 4. We condition on the event $\mathcal{E}_0$ described in Lemma 7. Then, we have some partial concept class $\mathcal{F}_G$ whose VC dimension is bounded by some distribution-dependent number $\hat{d}$ and has the property that $\forall \ell \in \mathbb{N}$, for $(x_1, y_1, \ldots, x_\ell, y_\ell) \sim \mathrm{P}^\ell$, $f(x_1, y_1, \ldots, x_\ell, y_\ell) = 1$ for some $f \in \mathcal{F}_G$. Thus, the conditions of Theorem 9 are satisfied and the conditional error rate of the output of our algorithm is $\mathbb{E}[\mathrm{er}(\hat{h}_n)|\mathcal{E}_0] \leq C'\hat{d}e^{-c'n/\hat{d}}$. Since $\mathcal{E}_0$ happens with probability at least $1 - \tilde{C}e^{-\tilde{c}n}$, we see that the unconditional error of the output is $\mathbb{E}[\mathrm{er}(\hat{h}_n)] \leq C'\hat{d}e^{-c'n/\hat{d}} + \tilde{C}e^{-\tilde{c}n} \leq Ce^{-cn}$, for some distribution-dependent constants $c, C > 0$. $\qquad\square$

### C.4 Slower than exponential is arbitrarily slow

In this section, we show that if the hypothesis class $\mathbb{H}$ admits an infinite VCL tree, then its learning rate is arbitrarily slow.

As with the exponential lower bounds, our result is rooted in a lower bound for lossy coding. Recall the definitions and notation for codes $(C, D)$ from Section B.1. However, in this case, we consider a slightly different scenario.

**Lemma 8.** *Let $\mathcal{D} = \mathcal{D}^*$ be the set of all binary strings of a given length $d \geq 128$. Let $\pi$ be the uniform distribution over $\mathcal{D}$. Let $\rho(x^*, x) = \frac{1}{d}\sum_{i=1}^{d} \mathbb{1}[x_i \neq x_i^*]$. Then, for any prefix-free code $(C, D)$ that achieves distortion $\mathbb{E}_{x^* \sim \pi}[\rho(x^*, D(C(x^*)))] \leq 1/8$, its rate must satisfy $\mathbb{E}_{x^* \sim \pi}[|C(x^*)|] > d/128$.*

*Proof.* Let $x^* \sim \pi$, and suppose $(C, D)$ is a prefix-free code with rate $\mathbb{E}[|C(x^*)|] \leq d/128$. By Markov's inequality, $\Pr(|C(x^*)| > d/32) \leq 1/4$. Consider the set $V = \{D(C(x)) : |C(x)| \leq d/32\}$. Note that, since $(C, D)$ is prefix-free, $|V| \leq 2^{d/32}$. For each $v \in V$, $\mathbb{E}[\rho(x^*, v)] = 1/2$, and thus a Chernoff bound implies $\Pr(\rho(x^*, v) \leq 1/4) \leq e^{-d/16}$. By the union bound,

$$\Pr(\exists v \in V : \rho(x^*, v) \leq 1/4) \leq |V|e^{-d/16} \leq 2^{d/32}e^{-d/16} \leq e^{-d/32} \leq e^{-2} < 1/4,$$

recalling that $d \geq 128$. Altogether, we have that

$$\mathbb{E}[\rho(x^*, D(C(x^*)))] \geq \frac{1}{4}\Pr(D(C(x^*)) \in V \text{ and } \nexists v \in V : \rho(x^*, v) \leq 1/4)$$

$$\geq \frac{1}{4}\left(\Pr(\nexists v \in V : \rho(x^*, v) \leq 1/4) - \Pr(D(C(x^*)) \notin V)\right)$$

$$\geq \frac{1}{4}\left(1 - \Pr(\exists v \in V : \rho(x^*, v) \leq 1/4) - \Pr(|C(x^*)| > d/32)\right)$$

$$> \frac{1}{4}\left(1 - \frac{1}{4} - \frac{1}{4}\right) = \frac{1}{8}.$$

This completes the proof. $\qquad\square$

Before presenting the result on arbitrarily slow rates, we will first need the following simple observation about the structure of VCL trees.

**Lemma 9.** *If $\mathbb{H}$ has an infinite VCL tree, then it also has an infinite VCL tree such that the points $x$ associated with any two nodes are disjoint: that is, $x_{\mathbf{y}_{\leq k}}^i \neq x_{\mathbf{y}'_{\leq k'}}^{i'}$ for $(i, \mathbf{y}_{\leq k}) \neq (i', \mathbf{y}'_{\leq k'})$ (in the notation of Definition 6).*

*Proof.* Consider any infinite VCL tree for $\mathbb{H}$. We will construct an infinite VCL tree having disjoint nodes by modifying the tree in a breadth-first way. We keep the root node as is. Then, inductively,

suppose we have already ensured that the first $n-1$ nodes in the breadth-first order are disjoint, and consider now the $n^{\text{th}}$ node. Let $k$ be the depth in the tree at which this node will appear. In the tree, as it currently exists at this point, we consider the subtree $T_n$ rooted at the current $n^{\text{th}}$ node in the breadth-first order. Let $N$ denote the number of points associated with the first $n-1$ nodes in the current tree. Since $N$ must be a finite number, we may select within the subtree $T_n$ a node with $N+k$ associated points (i.e., a node at depth $N+k$ in the current tree). Associated with this node, there must exist some $k$ points that are not associated with any of the $n-1$ previous nodes. We update the tree by replacing the $n^{\text{th}}$ node in the tree with a node with these $k$ points as its associated points. To define the subtree rooted at this node, for each of the $2^k$ classifications of these $k$ points, we choose one of the $2^N$ subtrees that were associated with this classification in the original node (from depth $N+k$). We repeat this for every node in each of these subtrees (in a breadth-first order), reducing the number of associated points to an arbitrary subset of the appropriate size (corresponding to its depth in this modified tree), and pruning all but one of its subtrees associated with each classification of the reduced set of points. Continuing this process inductively, we construct the infinite tree that never associates a point with more than one node. $\qquad\square$

We are now ready to prove that classes with an infinite VCL tree require arbitrarily slow rates.

**Theorem 11.** *If $\mathbb{H}$ has an infinite VCL tree, then $\mathbb{H}$ is interactively learnable with arbitrarily slow rates. This holds even if $P_{\mathcal{X}}$ is known to the learner.*

*Proof.* Suppose $\mathbb{H}$ has an infinite VCL tree. Let $\phi(x)$ be any positive decreasing bijection $[1,\infty) \to (0,1]$ with $\phi(x) \to 0$ as $x \to \infty$. We will show that the learning rate can be lower bounded by $\phi(256n)$, for infinitely many $n \in \mathbb{N}$. Since the rate of $\phi(n) \to 0$ may be chosen as slow as we like, this will establishes the result. For each $i \in \mathbb{N}$, we let $N_i = \lfloor \phi^{-1}(2^{-i-3}) \rfloor$; for simplicity, suppose $N_1 \geq 256$ and that $N_i$ is strictly increasing in $i$ (both of which are satisfied as long as $\phi(n)$ shrinks slowly enough in $n$). Also, suppose $\phi(x) \leq 2\phi(2x)$, which again would be satisfied as long as $\phi(x)$ shrinks sufficiently slowly in $x$ (i.e., slower than $1/x$).

We first describe the marginal distribution $P_{\mathcal{X}}$ on $\mathcal{X}$. The distribution is supported on points in a subtree of the infinite VCL tree of $\mathbb{H}$. By Lemma 9, we may assume, without loss of generality, that any point $x$ is associated with at most one node in the infinite VCL tree. We construct a subtree of this tree in a breadth-first manner. Set as the root node any node in the tree at depth $N_1$: that is, a node consisting of $N_1$ elements. Then, inductively, asssume that some leaf node $v$ in the construction-so-far has $k$ points associated with it; $v$ will always be a node chosen from the original tree. We need to attach $2^k$ different children to $v$ that correspond to all the possible classifications of the points of this node. We add these nodes from the corresponding $2^k$ subtrees rooted at the $2^k$ children of $v$ in the original tree, one from each subtree, so that they are valid extensions. Specifically, for each of the possible classifications, if the node we are adding for the corresponding branch will be the $i^{\text{th}}$ node added to the tree under-construction so far (in total), then we choose it to be any node in the corresponding subtree (of the original tree) at depth $N_i$: that is, a node with $N_i$ associated points. We add the nodes to the tree in this way, inductively, in a breadth-first manner: i.e., we attach all of the children to each node in a given layer before moving on to repeat this process on the next layer (which we have just finished adding). This completes the inductive construction of the special subtree, which will be the support of the distribution $P_{\mathcal{X}}$. Finally, to define $P_{\mathcal{X}}$, for the $i^{\text{th}}$ node in the breadth-first order of this subtree (i.e., the $i^{\text{th}}$ node added in the above construction), we assign $2^{-i}/N_i$ probability mass to each point $x$ associated with this node. Since the sets of points associated with nodes are disjoint across nodes, this assignment is well-defined. Moreover, since the $i^{\text{th}}$ node has $N_i$ associated points, this defines a probability measure on $\mathcal{X}$.

Now let us define the target labeling. For every infinite path from the root of the subtree constructed above, let us define a classification of all points associated with nodes in the subtree. The points $x \in \mathcal{X}$ associated with nodes on the path will take the classification implied by the path, which is well-defined since each branch from each node corresponds to a classification of all points associated with that node. For points in the subtree that are not found in nodes on the path, we define their classification (corresponding to this path) in a breadth-first manner, as follows. We let $h_i$ be a classifier in $\mathbb{H}$ that realizes the classifications of all points associated with nodes on the chosen path, up to depth $i$; such an $h_i$ exists, since there are a finite number of such points, and the paths are finitely realizable by definition (since they are subsets of paths in the original infinite VCL tree). We also let $V = \{h_i : i \in \mathbb{N}\}$. We inductively follow a breadth-first traversal (left-to-right order of the

nodes, and some order of points within each node); consider the next point $x$ in this traversal. To determine the label we assign (corresponding to the chosen path), if there are infinitely many $h \in V$ with $h(x) = 0$, we set the label of $x$ to 0, and otherwise we set its label to be 1. We update $V$ by discarding the classifiers that disagree with the label we assign. We then continue on to the next $x$ in the breadth-first traversal. Notice that by construction, $V$ initially contains an infinite number of functions, so at every $x$, either infinitely many $h \in V$ have $h(x) = 0$ or infinitely many $h \in V$ have $h(x) = 1$; either way, we assign a label that maintains the invariant that $V$ remains infinite after pruning the inconsistent functions. By induction, for every path from the root, this defines a corresponding classification of the entire subtree. Moreover, since the support of $P_\mathcal{X}$ is contained in the tree, and since $V$ is always non-empty, even after constraining to agree with the above labels on any finite number of points in the breadth-first order, we find that $P_\mathcal{X}$ together with the specified classification specify a realizable distribution on $\mathcal{X} \times \{0, 1\}$.

We pick the target labeling via the probabilistic method, choosing a path in the constructed subtree uniformly at random, and classifying the points associated with the subtree according to the corresponding classification, as defined above. Moreover, by the definition of queries in this work, we know there exists a $\theta^* \in \Theta^*$ for which $h(\theta^*, \cdot)$ takes these $h^*$ target classifications of points in the support; thus, we may take this $\theta^*$ value to define $h^* = h(\theta^*, \cdot)$, and the learner's queries will receive responses consistent with this $\theta^*$.

Notice that along the target path in this subtree, all nodes have some number of points $N_i$, whose labels are conditionally uniform random (given the event that the target path passes through that node). Hence, conditional on the target path passing through the $i^{\text{th}}$ node, we may regard the target classification of the associated $N_i$ points as a uniform random bit string of length $N_i$. Fix any learning algorithm $\hat{h}_n$. Let $H(s_{N_i}, \hat{s}_{N_i})$ be the Hamming distance of two bit strings $s_{N_i}, \hat{s}_{N_i}$ that correspond to the target labels and predicted labels (by $\hat{h}_n$), respectively, of the $N_i$ points associated with the $i^{\text{th}}$ node in the subtree in its breadth-first order. Then, based on the definition of $P_\mathcal{X}$, we know that any classifier $\hat{h}_n$ has $\mathrm{er}(\hat{h}_n) \geq \frac{2^{-i}}{N_i} H(s_{N_i}, \hat{s}_{N_i})$. Consider $n = \lceil N_i/256 \rceil$, which is strictly less than $N_i/128$ since $N_i \geq 256$. By Lemma 8, conditioned on the unlabeled data and internal randomness of the learner (if any) and the event that the target path passes through the $i^{\text{th}}$ node in the breadth-first order of the subtree, the conditional expectation of $H(s_{N_i}, \hat{s}_{N_i})$ is greater than $N_i/8$, so that for $X \sim P_\mathcal{X}$ independent of the data, learner, and random draw of the target labeling $h^*$, the conditional probability that $X$ is among the $N_i$ points associated with this $i^{\text{th}}$ node *and* $\hat{h}_n(X) \neq h^*(X)$ is greater than $2^{-i-3} \geq \phi(N_i) \geq \phi(256n)$, by the definition of $N_i$ and by monotonicity of $\phi$. Moreover, by the law of total expectation, the same is true if we average over the internal randomness of the learner: that is, we only condition on the event that the target path passes through the $i^{\text{th}}$ node of the subtree.

So far we have proven a lower bound on the conditional expected error of any algorithm, assuming the target path passes through a node that has roughly $256n$ points. However, for any fixed $n$, the probability that the target path passes through such a node can be very small. But notice that, to show the lower bound, we merely need to argue that there exists a (realizable) target labeling for which $\mathbb{E}[\mathrm{er}(\hat{h}_n)] \geq \phi(256n)$ for infinitely many $n$, which would be implied by showing

$$\mathbb{E}\left[\limsup_{n \to \infty} \frac{\mathbb{E}\left[\mathrm{er}\left(\hat{h}_n\right) \big| h^*\right]}{\phi(256n)}\right] \geq 1$$

when $h^*$ is the target labeling sampled randomly as described above.

Let $V_{d,h^*}$ be the set of (unlabeled) points $x$ associated with the node at depth $d$ of the target path, let $|V_{d,h^*}|$ be the size of that node, and define $n_{d,h^*} = \lceil |V_{d,h^*}|/256 \rceil$. We have that

$$\mathbb{E}\left[\limsup_{n \to \infty} \frac{\mathbb{E}\left[\mathrm{er}\left(\hat{h}_n\right) | h^*\right]}{\phi(256n)}\right] \geq \mathbb{E}\left[\limsup_{d \to \infty} \frac{\mathbb{E}\left[\mathrm{er}\left(\hat{h}_{n_{d,h^*}}\right) | h^*\right]}{\phi(256 n_{d,h^*})}\right]$$

$$\geq \mathbb{E}\left[\limsup_{d \to \infty} \frac{\mathrm{Pr}\left(\hat{h}_{n_{d,h^*}}(X) \neq h^*(X) \wedge X \in V_{d,h^*} | h^*\right)}{\phi(256 n_{d,h^*})}\right],$$

where $X$ is an independent $P_{\mathcal{X}}$-distributed point. Now note that

$$\Pr\left(\hat{h}_{n_{d,h*}}(X) \neq h^*(X) \wedge X \in V_{d,h*} \big| h^*\right) \leq \Pr\left(X \in V_{d,h*} \big| h^*\right) \leq 16\phi(256n_{d,h^*})$$

where the rightmost inequality holds due to the fact that $\Pr\left(X \in V_{d,h*} \big| h^*\right) = 2^{-i}$ for some $i$, and hence, since $N_i = \lfloor \phi^{-1}(2^{-i-3}) \rfloor$ and $\phi$ is decreasing, and $n_{d,h^*} = \lceil N_i/256 \rceil$, we have $2^{-i} \leq 8\phi(N_i) \leq 8\phi(128n_{d,h^*}) \leq 16\phi(256n_{d,h^*})$ (where the second inequality is due to $N_i \geq 256$ implying $n_{d,h^*} \leq N_i/128$, and the final inequality is due to $\phi(x) \leq 2\phi(2x)$).

Thus, we have shown that the ratio in the $\limsup$ is bounded, and therefore, Fatou's lemma implies that

$$\mathbb{E}\left[\limsup_{d\to\infty} \frac{\Pr\left(\hat{h}_{n_{d,h*}}(X) \neq h^*(X) \wedge X \in V_{d,h*} \big| h^*\right)}{\phi(256n_{d,h*})}\right]$$

$$\geq \limsup_{d\to\infty} \mathbb{E}\left[\frac{\Pr\left(\hat{h}_{n_{d,h*}}(X) \neq h^*(X) \wedge X \in V_{d,h*} \big| h^*\right)}{\phi(256n_{d,h*})}\right]$$

$$= \limsup_{d\to\infty} \mathbb{E}\left[\mathbb{E}\left[\frac{\Pr\left(\hat{h}_{n_{d,h*}}(X) \neq h^*(X) \wedge X \in V_{d,h*} \big| h^*\right)}{\phi(256n_{d,h*})} \bigg| V_{d,h^*}\right]\right]$$

$$= \limsup_{d\to\infty} \mathbb{E}\left[\frac{\Pr\left(\hat{h}_{n_{d,h*}}(X) \neq h^*(X) \wedge X \in V_{d,h*} \big| V_{d,h^*}\right)}{\phi(256n_{d,h*})}\right].$$

We have already argued above that

$$\Pr\left(\hat{h}_{n_{d,h*}}(X) \neq h^*(X) \wedge X \in V_{d,h*} \big| V_{d,h^*}\right) \geq \phi(256n_{d,h^*}).$$

Altogether, we have that

$$\mathbb{E}\left[\limsup_{n\to\infty} \frac{\mathbb{E}\left[\mathrm{er}\left(\hat{h}_n\right) \big| h^*\right]}{\phi(256n)}\right] \geq 1.$$

In particular, this implies that for any learning algorithm $\hat{h}_n$, there exists a deterministic choice of target labeling $h^*$ such that $\mathbb{E}\left[\mathrm{er}\left(\hat{h}_n\right)\right] \geq \phi(256n)$ for infinitely many $n \in \mathbb{N}$. This concludes the proof. $\qquad\square$