# OpenReview forum: "Universal Rates for Interactive Learning"
_NeurIPS.cc/2022/Conference — NeurIPS 2022 Accept_

### Official Review · Reviewer_4rao · 2022-07-01

**Rating:** 8
**Confidence:** 3
**Soundness:** 4 excellent
**Presentation:** 3 good
**Contribution:** 3 good

**Summary:**

In this paper, the universal learning framework is studied. In universal learning, unlike in uniform learning, the goal is to obtain decay rates for learning curves where the constants are potentially distribution-dependent. This can lead to improvement as compared to the distribution-independent formulation of uniform bounds. In contrast to previous work on universal learning, which focused on standard passive learning, this paper considers interactive learning, where the learner is allowed to send binary queries to an oracle. This includes the active learning setting, where the learner requests labels of unlabelled data. For this setting, three regimes of learning rates are shown, which give potentially dramatic improvements as compared to passive learning results for universal learning.

**Questions:**

The algorithms used in the proofs seem quite impractical. Do the presented results and conclusions have any concrete implications? Is it practically feasible to achieve the faster described rates with any computationally efficient algorithm?
What are the precise technical differences between your proofs and those of [BHM+21] for the passive setting?
How vital is the assumption of realizability? Is this a technical simplification, or do the results completely break down in its absence?
What are the remaining questions in this line of work?


Minor comments:
Line 176 "and and"
Missing space on line 342

**Limitations:**

The authors adequately discuss the limitations. Possibly, the practical considerations of the limitations of realizability, iid, and access to the unlabelled data distribution can be discussed further.

**Strengths And Weaknesses:**

The paper is overall very well-written, and gives significant results in a pertinent area. While the extension from passive to interactive is in a sense straight-forward, the technical details are non-trivial and the proofs use new algorithmic ideas. While I did not go through the proofs in detail, everything seems sound and well-written based on a skim reading. The main body of the paper does a commendable effort in trying to convey the intuition of the proofs and results.


In terms of weaknesses, the paper could benefit from a discussion of practical aspects. The algorithms used in the proofs seems highly impractical in terms of actual learning tasks due to the computational requirements, such as running learning algorithms on all possible classifications of a set of unlabelled data. The intuitive description of the algorithms are still not entirely easy to grasp, but I understand if it is hard to simplify the presentation more.

---

> ### Author Response · Authors · 2022-08-02
> **Response to Reviewer 4rao (Part 1)**
>
> We would like to thank the reviewer for finding our paper very well-written and our results significant. We will try to give more intuition about our algorithms and make them easier to grasp.
>
> > *The algorithms used in the proofs seem quite impractical. Do the presented results and conclusions have any concrete implications? Is it practically feasible to achieve the faster described rates with any computationally efficient algorithm?*
>
> We would like to first address the comment regarding the practicality of our algorithms. Notice that our main result characterizes the query complexity of interactive universal learning and is general in the sense that it applies to every class without assuming any structure on it. Thus, inevitably, the algorithm we use is very abstract. We hope and believe that for specific natural classes it will be possible to find explicit and efficient algorithms, and leave this as an important direction for future research. Relatedly, for specific natural classes and distributions we believe one will be able to get explicit bounds on the query and sample complexities. We also believe that our approaches will inspire practitioners to come up with heuristics that perform better in practice compared to the state of the art. We will add a discussion to the next version of our manuscript.
>
> > *How vital is the assumption of realizability? Is this a technical simplification, or do the results completely break down in its absence?*
>
> The techniques certainly rely on realizability, and we do not believe the same rates would hold for the agnostic setting.  Nevertheless, our work serves as a starting point for understanding the agnostic setting in future work.
>
> > *What are the remaining questions in this line of work?*
>
> We believe that there are a lot of interesting questions that remain open in this line of work. These include characterizing the unlabeled sample complexity (i.e. is it possible to achieve optimal rates both in terms of the number of queries and the number of samples?), designing efficient and practical algorithms for natural classes, and studying natural types of interactions such as relative queries (e.g. comparison queries). Moreover, the question you raised about the agnostic setting is also important.

---

> > ### Author Response · Authors · 2022-08-02
> > **Response to Reviewer 4rao (Part 2)**
> >
> > > *What are the precise technical differences between your proofs and those of [BHM+21] for the passive setting?*
> >
> > We now explain the technical differences between our work and [BHM+21] in detail.
> >
> > * Upper bound (no infinite Littlestone tree): We use the extension of the SOA that guarantees a finite number of mistakes when the learner is playing against an adversary who uses a class $H$ with only finite Littlestone trees. This was developed in [BHM+21], but the reader does not need to be familiar with the implementation of the algorithm – we treat it as a black box. We utilize this subroutine to develop a novel *active* learning algorithm, which uses only label queries, and achieves arbitrarily fast learning rate. Both the algorithm and its analysis are very different compared to the ones that appeared in [BHM+21].
> >
> > * Upper bound (no infinite VCL tree): The first step in our approach is the same as in [BHM+21], i.e. we use the Gale-Stewart game that was developed in [BHM+21] to end up with partial concept classes that have a bounded VC dimension. We utilize some results from [BHM+21] which show that the partial concept classes that correspond to batches of a (labeled) dataset satisfy some important properties (see Lemma 5). Then, we aggregate these partial classes into a ``majority’’ class (see Lemma 7). From that point forward, our approach diverges from the one in [BHM+21]. We develop a novel interactive learning algorithm which uses binary queries on subsets of the unlabeled data in order to figure out the correct labels of, roughly, $2^n$ points using $O(n)$ such queries. Then, we feed these points to the one-inclusion graph predictor that guarantees linear error in the number of points. Since the points are exponential in $n$, we get the result. In [BHM+21] the authors feed the labeled dataset of $n$ points that they have access to directly to the one-inclusion graph predictor, which gives an error rate that is linear in $n$.
> >
> > * Lower bounds: Both of the lower bounds we present are technically more challenging to establish than the ones in [BHM+21]. This challenge stems from the fact that our bounds hold in the case where the learner has *full* access to the marginal distribution $P_{X}$. Thus, the instances which give rise to the lower bounds are more complicated compared to their counterparts in the passive setting. Moreover, our proofs utilize a connection between learning algorithms and coding schemes with low distortion rate, which was not necessary in the proofs in [BHM+21]. For example, to show the lower bound in the case where $H$ has an infinite Littlestone tree, [BHM+21] consider a random path in the infinite given tree, then they define the marginal distribution $P_X$ to be supported only on that path and the target labels are determined by the path. However, in our setting this approach does not work because we show a lower bound in the case where the learner has full access to $P_X$. Our first step is to pre-process the tree and make sure that all the nodes have a different $x$-instance associated with them. Then, we define a distribution $P_X$ that is supported on every node of the tree. We also pick a random path and the labels of the nodes that are on the path are also determined by this choice. The next step is to pick the labels of the nodes that are not on the path in a consistent way. The last step is to establish a connection between coding schemes with low distortion rate and interactive learning algorithms and leverage some information-theoretic lower bounds in order to show the desired lower bounds on the learning rates. There are similar technical difficulties that we need to overcome in the case of an infinite VCL tree.
> > We will add this explanation to the next version of our manuscript.

---

### Official Review · Reviewer_gBuw · 2022-07-10

**Rating:** 7
**Confidence:** 4
**Soundness:** 4 excellent
**Presentation:** 4 excellent
**Contribution:** 3 good

**Summary:**

The authors study interactive learning in [BHMYH21]’s, universal learning model. Universal learning studies the asymptotic error rate of classification in a distribution-dependent fashion, in the sense that a class $H$ is learnable at rate $R(n)$ if for every (realizable) joint distribution over the data there exist constants $c$,$C$ such that $\mathbb{E}[err(h_n)] \leq CR(cn)$. This better models practical ML settings where a target distribution is fixed before the number of samples is chosen. In the interactive setting, the algorithm is allowed to ask arbitrary binary queries  (the learner may also use a finite but unbounded number of unlabeled samples), and $n$ denotes the number of queries rather than number of samples.

The authors show that interactive learning in the universal setting is characterized by the same trifecta as passive learning, but with strictly improved rates. Namely, they show that a hypothesis class satisfies exactly one of the three following

1. Learnable at arbitrarily fast rates
2. Learnable at exponential rate
3. Learnable at arbitrarily slow rates

Furthermore, they give a combinatorial characterization of which classes fall into these categories: classes with no infinite Littlestone tree satisfy 1, classes with no infinite VCL tree satisfy 2, an class with an infinite VCL tree satisfy 3. [BHMYH21] showed the same trifecta for passive learning, where the rate of 1 is exponential, 2 is linear, and 3 is arbitrarily slow. Thus the authors in this work fully resolve when and by how much interaction helps in the universal setting.

**Questions:**

Is it the case that the rate of error decay with respect to unlabeled data is substantially worse than in its unlabeled counterpart? If so is it known this is necessary to the characterization?

**Limitations:**

Yes

**Strengths And Weaknesses:**

This paper resolves the problem of characterizing interactive learning with arbitrary binary queries in the universal setting (a question in a sense first proposed in [BHW08]). This is an important theoretical contribution to the field of active learning and the study of the universal model. The techniques in the work are a mix of strategies developed in [BHMYH21]’s original study of passive universal learning (e.g. analysis of Gale-Stewart games) and novel algorithmic techniques such as reduction to interactive partial learning. The paper is well-written and the authors do a nice job of fitting their work into the literature (though it might be nice to also mention Long’s original work on partial learning “On agnostic learning with {0,*,1}-valued and real-valued hypotheses”). The work is likely to be of strong interest to the (theoretical) active learning community and of fair interest to the learning theory community at large.

The main weakness of the work lies in the very strong model of interactivity it considers. Namely the work makes two main assumptions that are atypically strong for the active learning literature:

1. The learner can make any binary query
2. The learner has access to an unbounded number of unlabeled samples

Learning with arbitrary binary queries is of theoretical interest, but is not particularly practically relevant. Even the membership query model (which allows arbitrary label queries) is too strong to be of interest in most applications. Similarly, it is typically assumed in active learning that the error rate with respect to unlabeled data also exhibits reasonable decay (in particular with a similar rate to what is enjoyed in the passive setting). Indeed in the uniform setting this can always be achieved. It is unclear from this work whether these issues are inherent and greatly affect the characterization of interactive universal learning, or merely artifacts of the proof.

---

> ### Author Response · Authors · 2022-08-02
> **Response to Reviewer gBuw**
>
> We would like to thank the reviewer for their conscientious reading of our work and for finding our paper well-written, our algorithmic techniques novel and the theoretical contributions important. We will add a short discussion about Long’s paper to the next version of our manuscript.
>
> > *1. The learner can make any binary query*
>
> We would like to first address the concern regarding the general interactive model we consider. We highlight that the algorithm which achieves arbitrarily fast rates requires only *label* queries on individual points, so it is captured by, arguably, the simplest interactive learning model, i.e. the active learning model. Moreover, the fact that the learner can use powerful queries makes our lower bounds particularly strong: in any natural interactive setting, the learner cannot achieve universal rate better than exponential if the class has an infinite Littlestone tree or better than arbitrarily slow if the class has an infinite VCL tree. We view our characterization as a first result in universal interactive learning and we hope and believe that it will lead to characterizations for more specialized models that allow more realistic types of queries.
>
> > *2. The learner has access to an unbounded number of unlabeled samples*
>
> We focussed on the query complexity for simplicity, as this is the main theme of this work, and of most works in the literature on interactive learning. This is inspired by the fact that unlabeled data are usually considered abundant, while the labels are more expensive.  For this reason, we also did not attempt to optimize the dependence on the number of unlabeled examples in the proofs.  That said, a careful inspection of the proofs reveals that for u unlabeled examples, when there is no infinite VCL tree, our rate in terms of u would be 1/u (matching the supervised rate, in contrast to the exponential rate in the number of queries).  When there is no infinite Littlestone tree, our proofs do no improve over this, and we hope this may be improved in future work
>
> > *Is it the case that the rate of error decay with respect to unlabeled data is substantially worse than in its unlabeled counterpart? If so is it known this is necessary to the characterization?*
>
> We kindly refer to the previous part of our response.

---

> > ### Comment · Reviewer_gBuw · 2022-08-05
> > **Thank you for Clarification**
> >
> > Thank you for the clarifying comments. These are indeed nice directions for future research.

---

### Official Review · Reviewer_whon · 2022-07-11

**Rating:** 9
**Confidence:** 3
**Soundness:** 4 excellent
**Presentation:** 4 excellent
**Contribution:** 4 excellent

**Summary:**

This work studies the universal learning rate for interactive learning, which, unlike the uniform learning setting that captures an upper envelope of the learning curves over families of distributions, concerns the best possible rate of convergence for every data distribution. It shows that interactive learning does have benefits over passive learning by showing much superior convergence rates in each category of the fundamental trichotomy. In addition, the bounds established can apply to arbitrary types of queries as long as they are binary.


**Questions:**

The paper is overall very well written. The only thing is that it may distinguish the techniques in this paper from the priors. The algorithmic designs are very clearly presented but might be hard for the reader to compare with the prior works, e.g. BHM+21, without reading them.


**Limitations:**

No concerns.


**Strengths And Weaknesses:**

This paper considers a very important problem (in the modern machine learning community) of interactive learning and provides insights through the lens of universal learnability which are quite interesting and inspiring. For example, just by a mild rearrangement of the quantifiers in its definition, the learning rates characterize the learnability of problems from a totally different perspective. The authors were able to show the benefit of interactive learning in two different categories, that is, the optimal learning rate for different distributions can be arbitrarily fast rates and exponential, while in the passive learning paradigm, the rates were exponential and polynomial respectively. The different categories appear to be closely related to the dimensionality of the infinite Littlestone tree and the VCL tree. The authors propose algorithms together with rigorous analysis with techniques built upon some tools from the prior works that study passive settings, fitting into the interactive learning scenario.

---

> ### Author Response · Authors · 2022-08-02
> **Response to Reviewer whon**
>
> > *The paper is overall very well written. The only thing is that it may distinguish the techniques in this paper from the priors. The algorithmic designs are very clearly presented but might be hard for the reader to compare with the prior works, e.g. BHM+21, without reading them.*
>
> We would like to thank the reviewer for their conscientious reading of our work and for finding our paper very well-written. In the following, we explain the differences in the techniques between our work and [BHM+21].
>
> * Upper bound (no infinite Littlestone tree): We use the extension of the SOA that guarantees a finite number of mistakes when the learner is playing against an adversary who uses a class $H$ with only finite Littlestone trees. This was developed in [BHM+21], but the reader does not need to be familiar with the implementation of the algorithm – we treat it as a black box. We utilize this subroutine to develop a novel *active* learning algorithm, which uses only label queries, and achieves arbitrarily fast learning rate. Both the algorithm and its analysis are very different compared to the ones that appeared in [BHM+21].
>
> * Upper bound (no infinite VCL tree): The first step in our approach is the same as in [BHM+21], i.e. we use the Gale-Stewart game that was developed in [BHM+21] to end up with partial concept classes that have a bounded VC dimension. We utilize some results from [BHM+21] which show that the partial concept classes that correspond to batches of a (labeled) dataset satisfy some important properties (see Lemma 5). Then, we aggregate these partial classes into a ``majority’’ class (see Lemma 7). From that point forward, our approach diverges from the one in [BHM+21]. We develop a novel interactive learning algorithm which uses binary queries on subsets of the unlabeled data in order to figure out the correct labels of, roughly, $2^n$ points using $O(n)$ such queries. Then, we feed these points to the one-inclusion graph predictor that guarantees linear error in the number of points. Since the points are exponential in $n$, we get the result. In [BHM+21] the authors feed the labeled dataset of $n$ points that they have access to directly to the one-inclusion graph predictor, which gives an error rate that is linear in $n$.
>
> * Lower bounds: Both of the lower bounds we present are technically more challenging to establish than the ones in [BHM+21]. This challenge stems from the fact that our bounds hold in the case where the learner has *full* access to the marginal distribution $P_{X}$. Thus, the instances which give rise to the lower bounds are more complicated compared to their counterparts in the passive setting. Moreover, our proofs utilize a connection between learning algorithms and coding schemes with low distortion rate, which was not necessary in the proofs in [BHM+21]. For example, to show the lower bound in the case where $H$ has an infinite Littlestone tree, [BHM+21] consider a random path in the infinite given tree, then they define the marginal distribution $P_X$ to be supported only on that path and the target labels are determined by the path. However, in our setting this approach does not work because we show a lower bound in the case where the learner has full access to $P_X$. Our first step is to pre-process the tree and make sure that all the nodes have a different $x$-instance associated with them. Then, we define a distribution $P_X$ that is supported on every node of the tree. We also pick a random path and the labels of the nodes that are on the path are also determined by this choice. The next step is to pick the labels of the nodes that are not on the path in a consistent way. The last step is to establish a connection between coding schemes with low distortion rate and interactive learning algorithms and leverage some information-theoretic lower bounds in order to show the desired lower bounds on the learning rates. There are similar technical difficulties that we need to overcome in the case of an infinite VCL tree.
> We will add this explanation to the next version of our manuscript.

---

### Author Response · Authors · 2022-08-02
**General Response to Reviewers**

We would like to thank all the reviewers for dedicating time to read the work and for their suggestions and positive feedback. Indeed, the reviews raise important questions such as characterizing the unlabeled sample complexity (i.e. is it possible to achieve optimal rates both in terms of the number of queries *and* the number of samples?), designing efficient and practical algorithms for natural classes,  and studying natural types of interactions such as relative queries (e.g. comparison queries).

These research directions are beyond the scope of the main result in this work and addressing them seems to require new ideas and techniques. We think that the questions the reviewers raise demonstrate the potential impact the current work can have in inspiring future research directions in interactive learning.

---

### Meta-Review · Area_Chair_Drx5 · 2022-08-25

**Recommendation:** Accept
**Confidence:** Certain

**Metareview:**

This paper provides a characterization of learning rates for interactive learning in the universal learning framework. All reviewers praised the novelty and quality of this submission.

**Award:**

Yes

---

### Decision · Program_Chairs · 2022-09-14

Accept